# Reward Learning through Ranking Mean Squared Error

Chaitanya Kharyal [* 1]   Calarina Muslimani [* 1]   Matthew E. Taylor [1 2]

## Abstract

Reward design remains a significant bottleneck in applying reinforcement learning (RL) to real-world problems. A popular alternative is reward learning, where reward functions are inferred from human feedback rather than manually specified. Recent work has proposed learning reward functions from human ratings rather than traditional binary preferences, enabling richer and potentially less cognitively demanding supervision. Building on this paradigm, we introduce a new rating-based RL method, Ranked Return Regression for RL (R4). At its core, R4 uses a novel ranking mean squared error loss that learns from a dataset of trajectory–rating pairs, treating the human-provided discrete ratings (e.g., "bad," "neutral," "good") as ordinal targets. Unlike prior rating-based approaches, R4 offers formal guarantees: its solution set is provably minimal and complete under mild assumptions. Empirically, using both human-provided and simulated ratings, we demonstrate that R4 consistently matches or outperforms existing rating and preference-based RL methods on robotic benchmarks from OpenAI Gym and the DeepMind Control Suite.

## 1. Introduction

Deep reinforcement learning (RL) has achieved remarkable success in games, where well-defined reward functions are available (Mnih et al., 2015; Schrittwieser et al., 2019). In contrast, real-world domains often lack clean specifications, making reward design a significant bottleneck to deploying RL in complex, practical applications (Knox et al., 2023). In practice, reward design is often an informal trial-and-error process where RL practitioners iteratively adjust a reward function until the RL agent exhibits acceptable behavior

*Equal contribution [1]Department of Computing Science, University of Alberta, Canada [2]Alberta Machine Intelligence Institute (Amii), Alberta, Canada. Correspondence to: Chaitanya Kharyal <kharyal@ualberta.ca>.

*Proceedings of the 43rd International Conference on Machine Learning*, Seoul, South Korea. PMLR 306, 2026. Copyright 2026 by the author(s).

(Booth et al., 2023). This procedure can be error-prone, resulting in reward misspecification where practitioners inadvertently define a reward function that does not align with the true task objective. This can result in the agent learning undesirable or unintended behaviors (Skalse et al., 2022; Pan et al., 2022). Notably, these challenges have been observed even in tabular domains, illustrating that reward design remains a core challenge (Muslimani et al., 2025).

A popular alternative to manual reward engineering is *reward learning*, where reward functions are inferred from human feedback rather than explicitly designed. This feedback can take various forms, including scalar evaluations (Knox & Stone, 2009; MacGlashan et al., 2017; Metz et al., 2025), demonstrations (Taylor, 2018; Arora & Doshi, 2021), or pairwise preferences over behaviors (Christiano et al., 2017). Of these approaches, preference-based RL (PbRL) has gained particular traction due to its low human effort and its role in large language models (OpenAI, 2023).

Despite its success, learning from binary preferences can be limiting. For one, each binary comparison conveys only a single bit of information, making reward learning sample inefficient in terms of required preference labels. As a result, more human time may be required compared to collecting multi-class ratings, increasing the overall feedback burden. Moreover, such feedback is inherently relative. It indicates which behavior is preferred, but not by how much, nor whether either option is good in an absolute sense. For example, if both behaviors under comparison are poor, a human can at best indicate that they are equally preferable, but cannot express that both are low-quality overall.

Recent work has introduced a new paradigm known as rating-based RL (RbRL) (White et al., 2024), in which humans provide discrete, multi-class ratings rather than binary preferences to guide reward learning. Instead of comparing two behaviors and selecting the preferred one, the human observes a single behavior at a time and assigns it a rating from a fixed scale. This shift enables the collection of richer feedback, as ratings can capture both relative and absolute assessments of trajectory quality. Furthermore, studies have found that humans experience less cognitive effort and greater task success when providing ratings, compared to expressing preferences (White et al., 2024).

The advantages of RbRL motivate the development of more

efficient algorithms for learning from ratings. To this end, we propose a new rating-based RL method: Ranked Return Regression for RL (R4). It learns reward functions from trajectories labeled with ordinal ratings using a novel ranking mean squared error loss. At each training step, we sample one trajectory per rating class, compute its predicted returns under the reward model, and rank them using a differentiable sorting operator (i.e., soft ranks). We then minimize a mean squared error loss between the resulting soft ranks and the human's ratings. Our contributions are as follows:

1. We propose Ranked Return Regression for RL, a rating-based RL algorithm that leverages a novel ranking mean squared error (rMSE) loss to train reward functions from trajectories labeled with ordinal ratings.

2. We establish that rMSE is the first rating-based RL objective with provable minimality and completeness guarantees under mild assumptions.

3. We validate R4 in both offline and online feedback settings using simulated feedback, demonstrating that it can outperform rating and preference-based RL algorithms across several robotic locomotion tasks.

4. We conduct an ethics-approved human-user study in which participants provide ratings for two robotic tasks, and find that R4 outperforms RbRL despite substantial inter-participant variability. Participants also report low cognitive workload when providing ratings, indicating that the rating modality is a cognitively feasible approach. Lastly, we open-source this dataset to support future research on rating-based reward learning.

Taken together, these contributions emphasize rating-based RL methods that are both theoretically grounded and effective in leveraging human feedback.

## 2. Related Work

Reward learning is a broad field in which reward functions are inferred from various forms of human feedback, including demonstrations, preferences, scalar evaluations, ratings, or combinations thereof. One approach is inverse RL, which learns reward functions from demonstrations (Ng & Russell, 2000; Brown et al., 2019). However, it is argued that providing demonstrations can be time-consuming and difficult (Akgun et al., 2012; Lee et al., 2021).

Other approaches rely on preference feedback, where human users typically provide binary preferences over pairs of behaviors (Christiano et al., 2017; Lee et al., 2021). This form of supervision has gained traction, as it is often considered more intuitive and less demanding for humans than providing full demonstrations. However, binary preferences

can be limited in the richness of information they convey. To address this issue, Wilde et al. (2021) explored scaled preferences, where users indicate not just which behavior they prefer, but also the strength of that preference (e.g., on a scale from "strongly prefer A" to "strongly prefer B"). These graded comparisons have been shown to outperform strict binary preferences, offering more informative supervision for reward learning.

Similarly, scalar feedback methods provide rich signals by allowing humans to rate behaviors directly. For example, the TAMER framework allows humans to provide binary signals indicating whether a behavior is considered optimal (Knox & Stone, 2009). Later, Cabi et al. (2020) introduced reward sketching, where, for a given behavior, humans continuously provide a scalar signal indicating progress toward a goal. Recent work on reward modeling for LLMs has also moved beyond binary preferences to use ordinal feedback (Liu et al., 2025a;b). Most similar to our approach is rating-based RL (White et al., 2024). It handles multi-class ratings by using a new cross-entropy loss. In this setup, humans assign class labels such as "good," "okay," or "bad" to trajectories, and these labels are then used to train a reward model.

In addition to the type of feedback, reward learning can also be categorized by how feedback is collected. In the offline setting, reward models are trained on static datasets of labeled trajectories (i.e., labeled with preferences, ratings, etc) and the learned reward is then used to train an RL agent (Shin et al., 2023). In the online setting, the agent interacts with the environment and receives feedback in real time; trajectories generated by the agent are periodically labeled, and the reward model is updated continuously as the agent learns its policy (Christiano et al., 2017).

## 3. Background

**Markov Decision Process Without Rewards (MDP\R),** is an MDP in which the reward function is unspecified (Abbeel & Ng, 2004). In this work, we consider an MDP\R augmented with human-provided ratings (White et al., 2024). Formally, it is defined as: $\mathcal{M} = (\mathcal{S}, \mathcal{A}, T, \rho, \gamma, n, \mathcal{D})$, where $\mathcal{S}$ and $\mathcal{A}$ are the state and action spaces, $T : \mathcal{S} \times \mathcal{A} \times \mathcal{S} \rightarrow [0, 1]$ is the transition probability function, $\rho$ is the initial state distribution, and $\gamma \in [0, 1)$ is the fixed (not learned) discount factor. In our setting, a human observes trajectory $\tau_i$, where $\tau_i = (s_1^i, a_1^i, \ldots, s_T^i, a_T^i)$, and assigns it a rating $c_i \in \{0, 1, \ldots, n - 1\}$, where $c_i$ indicates the perceived quality of the trajectory. A rating of $0$ represents the lowest quality, while $n - 1$ represents the highest. Let $c(\tau_i)$ be a function that maps the trajectory $\tau_i$ to the corresponding rating class. Note that the rating classes can also be assigned descriptive labels to aid interpretation. For instance, with $n = 3$ rating classes, the labels might be: 0 ("bad"), 1 ("neutral"), and 2 ("good"). This rating process is repeated for all

trajectories, and the resulting data is grouped by rating class. Specifically, for each rating class $k \in \{0, 1, \ldots, n-1\}$, we define a subset $\mathcal{D}_k = \{(\tau_i, k) \mid c(\tau_i) = k\}$ containing all trajectories assigned to rating class $k$. The complete dataset is then $\mathcal{D} = \bigcup_{k=0}^{n-1} \mathcal{D}_k$ with $|D| = K$.

In the standard RL setting, the MDP includes a reward function $r : \mathcal{S} \times \mathcal{A} \to \mathbb{R}$, which provides a numerical reward for each state-action pair. The objective is to find an optimal policy $\pi^*$ that maximizes the expected discounted return, $G_r$ (with respect to reward $r$) defined as: $\mathbb{E}_\pi \left[ \sum_t \gamma^t r(s_t, a_t) \right]$. In contrast, in an MDP\R with human ratings, no reward function is available; instead, the human rating components (e.g., $n$, $\mathcal{D}$) serve as the sole source of feedback. The goal then is two-fold: (1) to learn a reward model $\hat{r}_\theta$, parametrized by $\theta$, from human-provided ratings; and (2) to learn a policy that maximizes the expected discounted return, $\hat{G}_\theta$, such that the resulting policy produces behaviors that satisfy the human's ratings.[1]

**Differentiable (Soft) Ranking** refers to a class of algorithms that make sorting and ranking operations differentiable (Grover et al., 2019; Blondel et al., 2020; Petersen et al., 2022). In R4, we use the algorithm proposed by Blondel et al. (2020), which assigns continuous, differentiable ranks to a set of values and includes a regularization hyperparameter: higher values produce smoother but less accurate ranks. For example, given the values $[3.2, 1.0, 4.5]$, it produces soft ranks $\hat{R} = [1.5, 0.7, 2.5]$, whereas the hard ranks are $R = [1, 0, 2]$, with the highest value ranked 2 and the lowest 0. The differentiable nature of soft ranks allows gradients to propagate through the ranking operation during optimization, which is not possible with hard ranks.

**Rating Based Reinforcement Learning** is a form of reward learning, where reward models are trained from discrete ratings via supervised learning, rather than from preferences (White et al., 2024). In particular, they introduced a cross-entropy–style loss function defined as:

$$\mathcal{L}_{\text{RbRL}} = - \sum_{\tau \in D} \left( \sum_{i=0}^{n-1} \mu_\tau(i) \log(Q_\tau(i)) \right),$$

where $\mu_\tau(i)$ is an indicator function that equals 1 if the human label for trajectory $\tau$ is class $i$, and 0 otherwise. Furthermore, the function $Q_\tau(i)$ is defined as:

$$Q_\tau(i) = \frac{e^{-k(\hat{G}_\theta(\tau) - B_i)(\hat{G}_\theta(\tau) - B_{i+1})}}{\sum_{j=0}^{n-1} e^{-k(\hat{G}_\theta(\tau) - B_j)(\hat{G}_\theta(\tau) - B_{j+1})}}.$$

Here, $\{B_i\}_{i=0}^{n}$ are class decision boundaries, $\hat{G}_\theta(i) \in [0, 1]$ denotes the normalized predicted return of trajectory $\tau$ un-

der the learned reward model $\hat{r}_\theta$ and $k$ is a hyperparameter. As reproduced in Appendix A.1, $\mathcal{L}_{\text{RbRL}}$ encourages the predicted returns of all trajectories within a class to concentrate around the midpoint $\frac{B_i + B_{i+1}}{2}$ (White et al., 2024).

# 4. Ranked Return Regression for RL – R4

Given a set of rating classes $c_0, \ldots, c_{n-1}$, where $i < j$ implies that trajectories in class $c_i$ are rated lower than those in class $c_j$, we assume that for any $\tau_a \in \mathcal{D}_i$ and $\tau_b \in \mathcal{D}_j$, the (unobserved) return under the the human's implicit reward function, $r^*$, satisfies $G^*(\tau_a) < G^*(\tau_b)$. Here, $G^*(\tau) = \sum_t \gamma^t r^*(s_t, a_t)$ denotes the discounted return of trajectory $\tau$ with respect to the human's reward function. While ratings assign trajectories independently to discrete classes, we can construct a ranking by ordering trajectories according to their assigned classes. Then, by sampling one trajectory from each class, we obtain a perfectly ordered ranking over $n$ trajectories (where $n$ is the number of rating classes).

We leverage this observation to define a novel ranking mean squared error (rMSE) objective over a set of trajectories, which serves as a supervised learning loss for training a reward function from trajectory ratings. We can then use this learned reward function, in place of an engineered reward, to train a policy to maximize the expected discounted return, $\hat{G}_\theta$ denoted as $\mathbb{E}_\pi \left[ \sum_t \gamma^t \hat{r}_\theta(s_t, a_t) \right]$. We refer to this rMSE-based training pipeline as Ranked Return Regression for RL. We demonstrate that this algorithm is flexible, applying it in both offline and online feedback settings.

## 4.1. Ranking Mean Squared Error Objective

To learn the reward function $\hat{r}_\theta$ from the dataset $\mathcal{D}$, we sample one trajectory $\tau_i$ from each class dataset $\mathcal{D}_k$. The discounted return for each trajectory is estimated as:

$$\hat{G}_\theta(\tau_i) = \sum_t \gamma^t \hat{r}_\theta(s_t, a_t)$$

We then rank the predicted discounted returns $\{\hat{G}_\theta(\tau_i)\}_{i=0}^{n-1}$ using a differentiable sorting algorithm (Blondel et al., 2020), producing a vector of soft ranks $\hat{R}$, where $\hat{R}(\tau_i)$ denotes the soft rank corresponding to trajectory $\tau_i$. The rMSE loss is computed as the mean squared error between the soft rank and the rating class provided by a human:

$$\mathcal{L}_{\text{rMSE}} = \frac{1}{n} \left[ \sum_{i=0}^{n-1} \left( \hat{R}(\tau_i) - c(\tau_i) \right)^2 \right] \tag{1}$$

For example, suppose the rating classes for the sampled trajectories are $c = [1, 2, 0]$, and the predicted soft ranks are $\hat{R} = [0.0, 2.0, 1.0]$. The rMSE loss is computed as:

$$\mathcal{L}_{\text{rMSE}} = \frac{1}{3} \left[ (0.0 - 1.0)^2 + (2.0 - 2.0)^2 + (1.0 - 0.0)^2 \right]$$

---

[1]Note that reward evaluation is not well defined in the literature; see Muslimani et al. (2025).

In this example, the predicted ranks for the first and third trajectories deviate from the corresponding rating. Since the soft ranks are differentiable with respect to the reward parameters, minimizing this loss allows the model to adjust $\hat{r}_\theta$ to better align with the ratings provided by a human. See Figure 1 for an overview of the R4 training process. Next, we outline several advantages of the rMSE loss over RbRL:

1. **Eliminating hyperparameters:** The RbRL objective requires specifying rating class boundaries $B$. A trajectory is assigned to class $k$ only if its return lies between $B_k$ and $B_{k+1}$. Our approach does not require such explicitly defined boundaries.

2. **Preserving within-class diversity:** The RbRL objective encourages all trajectories in a class to have predicted returns close to the midpoint $\frac{B_k + B_{k+1}}{2}$, thereby ignoring intra-class diversity. In contrast, the rMSE objective does not enforce such a constraint, allowing greater flexibility in modeling returns within the same class. In Proposition 1, we show that enforcing such constraints on intra-class variability can be detrimental.

3. **Dynamic rating classes:** The rMSE objective allows the number and structure of rating classes to change dynamically during training. This flexibility means raters are not restricted to a fixed rating scheme; they can introduce new classes if they want finer distinctions or merge existing ones when coarser ratings are more natural. In contrast, extending the RbRL objective to dynamic classes is non-trivial, as its performance can degrade when the number of bins deviates from the optimal range (White et al., 2024).[2]

### 4.2. Design Choices for the Online Feedback Setting

For the online feedback experiments, we implement three strategies to use human feedback more effectively. First, we apply a *dynamic feedback schedule* that collects feedback more frequently at the start of training and gradually reduces the feedback frequency as training progresses. Next, to determine which trajectory segments to sample, we use a *stratified sampling approach*. Specifically, we maintain a dataset of the 50 most recent trajectories and select a fraction from high-predicted-return trajectories, with the remainder drawn from lower-predicted-return ones. For each selected trajectory, we extract a sub-trajectory of fixed length by choosing either a random segment or the segment with the highest predicted return, each with equal probability. This heuristic aims to balance exploration of diverse trajectories with attention to promising ones. Further details about the preference collection schedule and the sampling strategy are provided in Appendix D.3. We test the impact of these strategies on learning progress in Section 6.4.

Moreover, in R4, we use *dynamic rating classes* to better accommodate how humans might provide feedback. Early in training, when the agent produces mostly low-quality behavior, we use finer-grained bins to distinguish poor trajectories, allowing humans to provide more informative feedback. As higher-quality trajectories emerge, these bins are merged into a single "low quality" class, and when a trajectory's return falls outside the current range, a new class is introduced. Both of these behaviors reflect the concepts of response shift and recalibration, where humans adjust their internal standards over time (Visser et al., 2000; 2005).

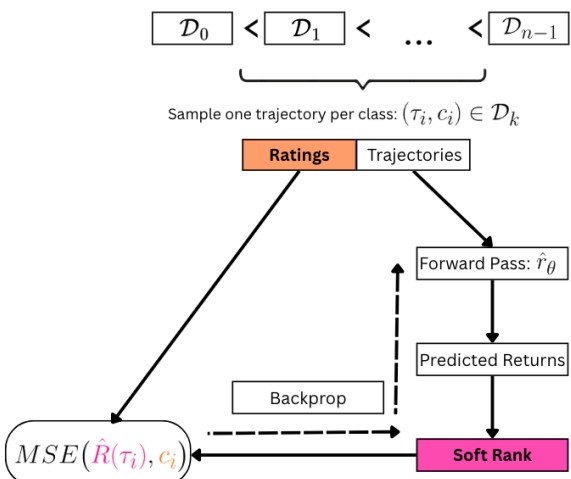

*Figure 1.* Illustration of the R4 training process: Given a dataset of trajectory–rating pairs, we compute the predicted return for each trajectory under $\hat{r}_\theta$ and apply a differentiable sorting algorithm to obtain soft ranks, $\hat{R}$. Then, we minimize the MSE between the soft ranks and the human ratings.

## 5. Theoretical Results

In this section, we present theoretical results to characterize the solution space of the rMSE objective.

**Assumption 1.** *The human has an unknown deterministic reward function $r^*$, which induces (unobservable) returns $G^*(\tau)$ for each trajectory $\tau$ in the dataset $\{\tau_i\}_{i=1}^K$. These trajectories are grouped into $n$ rating classes $\{c_j\}_{j=0}^{n-1}$ in a noise-free manner, with each class corresponding to a range of returns under $G^*$.*

**Assumption 2.** *$r^*$ belongs to the hypothesis class.*[3]

**Assumption 3.** *The differentiable ranking function used by rMSE produces the hard rank of each element in a list.*

Assumption 1 is inherent to our problem formulation and common in theoretical analyses of PbRL (Song et al., 2024). Assumption 2 represents a standard requirement in the optimization literature. For Assumption 3, we demonstrate in

---

[2]We confirm this performance degradation in Appendix C.

[3]The theory results are not limited to a specific function class.

Appendix A.4 that it holds in practice via the differentiable sorting method of Blondel et al. (2020). However, the requirement for differentiability is only a byproduct of using a gradient-based optimizer; *it is not a constraint of the objective itself*, which could otherwise be optimized using hard rankings with non-gradient-based methods. Nonetheless, we later *relax this assumption entirely*.

**Definition 1.** *The set of reward functions $\mathcal{R}$ is the set of feasible solutions that satisfy Assumptions 1–2. More formally,*

$$\mathcal{R} \triangleq \{\hat{r}_\theta \mid c(\tau_i) < c(\tau_j) \implies \hat{G}_\theta(\tau_i) < \hat{G}_\theta(\tau_j), \\ \forall \tau_i, \tau_j \in \mathcal{D}\}. \quad (2)$$

Note that $\mathcal{R}$ is the set of reward functions in the hypothesis class that we care to find, as they satisfy all assumptions imposed by the problem.

**Proposition 1.** *Under Assumptions 1–3, the human's reward function $r^*$ is always contained in the solution set of the rMSE objective, but is not guaranteed to be in the solution class of RbRL objective*

$$r^* \in \arg\min_\theta \mathcal{L}_{rMSE}(\theta), \text{ but} \\ r^* \notin \arg\min_\theta \mathcal{L}_{RbRL}(\theta) \text{ in general.} \quad (3)$$

The proof is given in Appendix A.2.1. Next, in Theorem 1 we show that the solution set of the rMSE objective (denoted $\mathcal{R}_{rMSE}$) is complete and minimal under Assumptions 1–3. This means the set of reward functions, $\mathcal{R}$, as specified in Definition 1 is equivalent to the rMSE solution set. In other words, there exists no other objective function that can further reduce the rMSE solution set without introducing additional assumptions. Doing so would risk excluding potential reward functions in $\mathcal{R}$. This also means that any reward function outside the rMSE solution set is not $r^*$.

**Theorem 1.** *Under Assumptions 1–3, the solution set of rMSE is complete and minimal. Formally,*

$$\hat{r}_\theta \in \mathcal{R} \iff \hat{r}_\theta \in \mathcal{R}_{rMSE}, \ \forall \hat{r}_\theta \quad (4)$$

The proof is given in Appendix A.2.2. Theorem 1 establishes the completeness and minimality of rMSE under Assumptions 1–3. To handle cases where Assumption 3 fails, we introduce a relaxed version of it (Assumption 4) and prove a corresponding result in Theorem 2; but empirically, Assumption 3 holds for a wide set of regularization strengths in Blondel et al. (2020) (see Appendix C, Figure 6).

**Assumption 4** (Relaxed Assumption 3). *For any element in the array $\mathbf{v}$, the rank predicted by the differentiable sorting operator, $\hat{R}$, differs from its hard rank, $R$, by at most $\epsilon$. Formally,*

$$\left|\hat{R}(v_i) - R(v_i)\right| \leq \epsilon, \ \forall i,$$

*where $0 \leq \epsilon < \frac{\sqrt{2n}-2}{n-2}$ is a constant and $n > 2$ is the number of elements in $\mathbf{v}$.*

**Definition 2.** *The relaxed solution set, $\widetilde{\mathcal{R}}_{rMSE}$, for $\mathcal{L}_{rMSE}$ is:*

$$\widetilde{\mathcal{R}}_{rMSE} \triangleq \{\hat{r}_\theta \mid \mathcal{L}_{rMSE}(\theta) \leq \epsilon^2\}.$$

**Theorem 2.** *Under Assumptions 1–2 and 4, $\widetilde{\mathcal{R}}_{rMSE}$ is complete and minimal. Formally,*

$$\hat{r}_\theta \in \mathcal{R} \iff \hat{r}_\theta \in \widetilde{\mathcal{R}}_{rMSE}, \ \forall \hat{r}_\theta \quad (5)$$

See proof in Appendix A.3. Theorem 2 shows that, as long as the error of the ranking operator satisfies the bound in Assumption 4, the guarantees with rMSE still hold.

## 6. Experimental Setup and Results

In this section, we first present the experimental design and results for the offline feedback setting, then those for the online feedback setting, and end with the human-user study.

### 6.1. Offline Feedback Experiments

**Experimental Setup.** We evaluate RbRL and R4 in the offline feedback setting in OpenAI Gym domains: `Reacher`, `Inverted Double Pendulum`, and `Half Cheetah`. In this setting, a reward model is trained on a static dataset of feedback, as opposed to the online feedback setting, where feedback is collected iteratively during RL training. The offline setup avoids additional hyperparameter choices such as the feedback frequency or trajectory sampling method, allowing us to better isolate the impact of the learning objective. While it also serves to show that R4 extends to offline settings, our primary goal is to demonstrate the performance gains that come specifically from replacing the RbRL loss with the R4 loss, rMSE.

To obtain offline trajectories, we train a Soft Actor-Critic (SAC) agent (Haarnoja et al., 2018) from Stable-Baselines3 (Raffin et al., 2021) using the environment reward function and store the resulting trajectories along with their environment returns. To systematically evaluate performance, we generate simulated ratings that assign scalar scores to each trajectory based on its environment return. Specifically, we define a set of return thresholds, where each trajectory is labeled according to the return bin it falls into, such that any trajectory $\tau_i$ with return $b[k] \leq G(\tau_i) < b[k+1]$ is assigned rating class $c(\tau_i) = k$. We then construct a balanced dataset by sampling an equal number of trajectories from each class, with $\mathcal{D}_k$ denoting the subset for class $k$ and $\mathcal{D} = \bigcup_{k=0}^{n-1} \mathcal{D}_k$.

Reward models are trained via supervised learning on $\mathcal{D}$, and we evaluate R4 against RbRL under the same training procedure. We also include the environment reward function as a baseline. Full details for reproducibility are provided in Appendix D.4. After training the reward model $\hat{r}_\theta$, we train

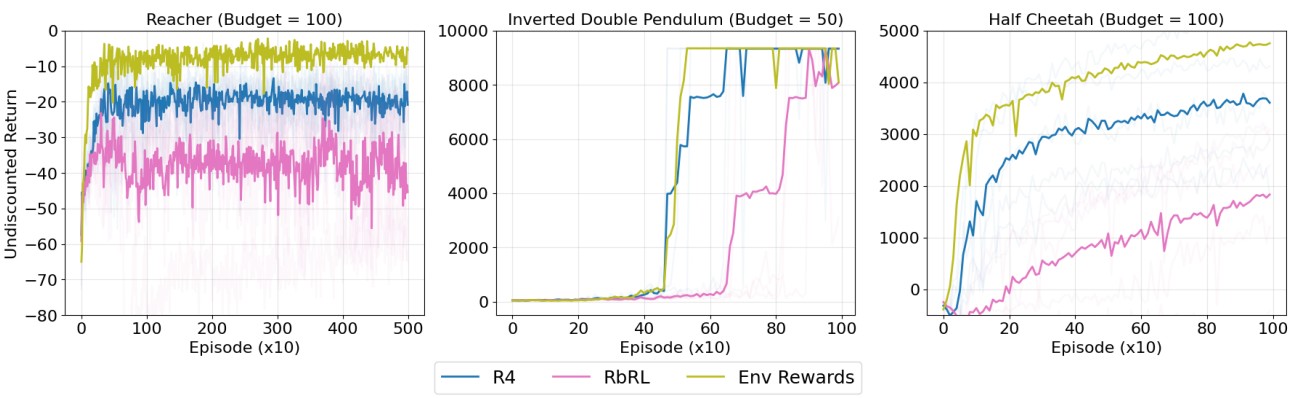

*Figure 2.* Performance of a SAC agent trained with (1) R4, (2) RbRL, and (3) the environment reward.

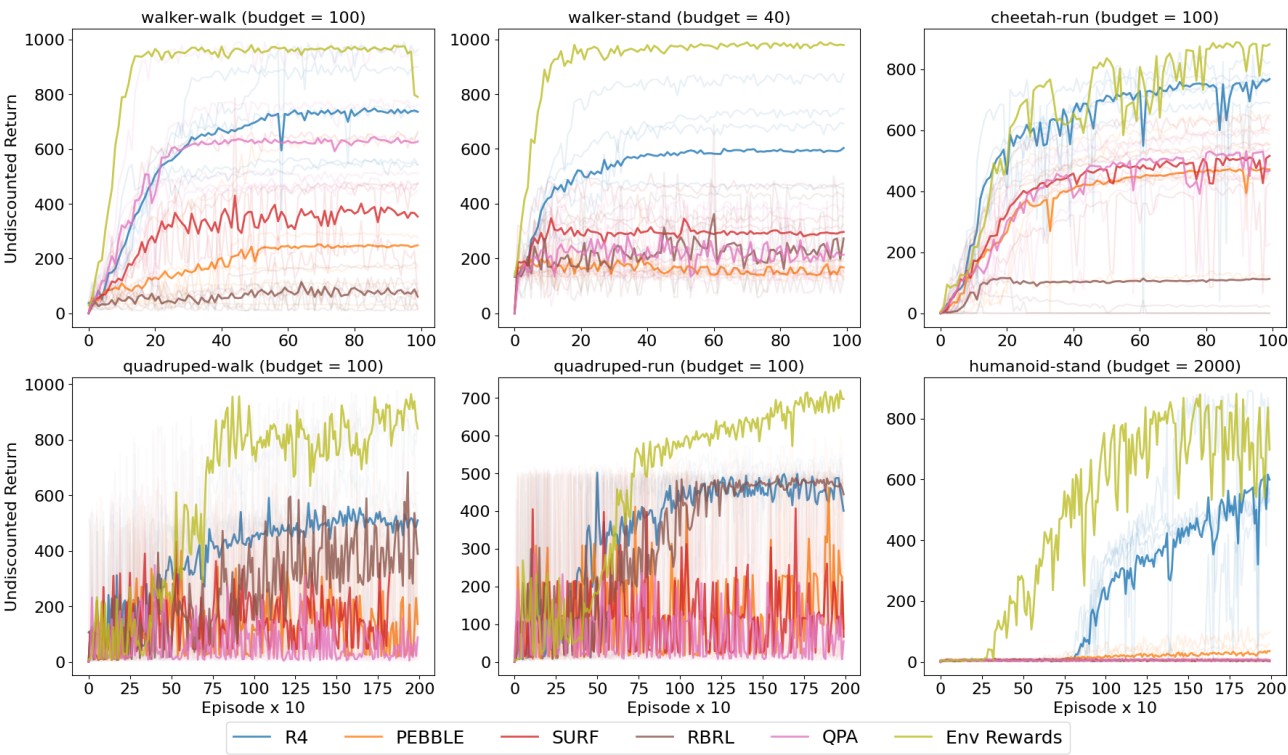

*Figure 3.* Online SAC training using the environment reward or rewards learned from different rating- and preference-based algorithms.

an RL agent on an unseen environment seed using $\hat{r}_\theta$ as the reward. This process is repeated for five random seeds to account for variability in both reward learning and policy optimization. Performance is measured using the environment reward. We report learning curves with individual runs shown in light colors and the mean in a darker color. To test for significant differences between R4 and RbRL, we perform $t$-tests with a significance level of $\alpha = 0.05$. See Appendix C.2 for detailed information on the $t$-test results.

**Offline Feedback Results.** We observe that under otherwise identical conditions, reward functions trained with

R4 consistently lead to better downstream RL performance than those learned with RbRL (see Figure 2). In particular, compared to RbRL, R4 reward functions led to either statistically faster SAC learning or higher final returns across all tested domains ($p < 0.03$).

### 6.2. Online Feedback Experiments

**Experimental Setup.** In our online experiments, a SAC agent interacts with the environment and learns from the learned reward function $\hat{r}_\theta$. From these interactions, we periodically sample trajectory segments and assign simulated ratings or preferences based on the en-

vironment's reward function. This feedback is then used to update the reward model, guiding the agent's future behavior. We evaluate R4 against RbRL and three preference learning algorithms: PEBBLE (Lee et al., 2021), SURF (Park et al., 2022), and QPA (Hu et al., 2024), across six DMC environments: `Walker-walk`, `Walker-stand`, `Cheetah-run`, `Quadruped-walk`, `Quadruped-run`, and `Humanoid-stand`. For all methods, we fix a feedback budget: in rating-based approaches, it counts rated trajectories, while in preference-based approaches, it counts pairwise comparisons. Since each comparison involves two trajectories, the same budget requires a human to assess twice as many trajectories in preference-based methods.

All implementation details are provided in Appendix D.3. For the online experiments, we follow the standard SAC implementation from PEBBLE (Lee et al., 2021). Regarding the reward learning components, the baselines use their default configurations: a uniform feedback schedule and their respective trajectory sampling strategies (uncertainty-based for all except QPA, which uses a near on-policy strategy). For completeness, we tested the baselines with our dynamic feedback schedule; however, this degraded performance (see Appendix C, Figure 23). Therefore, we report results using each method's strongest configuration. As in the prior section, performance is measured using the (unobserved) environment reward function, with learning curves showing five individual runs (light) and their mean (dark). $t$-tests ($\alpha = 0.05$) with Bonferroni correction identify statistically significant differences between R4 and other methods. As we conduct four comparisons per domain, we use a corrected significance threshold of $\frac{\alpha}{4} = 0.0125$ to control the family-wise error rate. Appendix C.2 has additional results.

**Online Feedback Results.** Figure 3 shows that R4 consistently matches or outperforms existing approaches across all tested environments. In particular, using R4, the SAC agent learns significantly faster than all baselines in three of the six environments and achieves higher final returns in four of the six environments ($p < 0.0125$).

## 6.3. Experiments with Human Ratings

Sections 6.1 and 6.2 evaluate the effectiveness of R4 using simulated feedback generated according to the environment reward function. While simulated experiments enable large-scale evaluation, they fail to capture key nuances of human interaction, as human feedback is often shaped by systematic biases (Ghosal et al., 2023). Moreover, testing with human participants is particularly vital given the current state of the field. A review of PbRL literature from 2012–2024 reveals a significant shortfall in human evaluation: less than 50% of proposed algorithms were validated with non-author human participants (Muslimani & Taylor, 2025). This gap

is alarming as using only simulated feedback risks producing models that fail to generalize to the inconsistencies and biases inherent in real human feedback.

We conducted an ethics-approved human-user study with 8 participants (4 male, 4 female). The study followed the offline feedback setting described in Section 6.1, in which participants provided ratings on a fixed dataset of trajectory segments. Participants provided 100 ratings in one or both domains: OpenAI Gym `Reacher` ($n_{\text{sample size}} = 7$) and `Hopper` ($n_{\text{sample size}} = 6$); with the ability to choose the number of rating classes used. See full details on the user interface and the human-rated dataset in Appendices B.1–B.2 and B.4, respectively. Using this feedback, we trained five reward models per participant. Each learned reward model was then used to train a separate SAC agent. Lightly colored curves in Figure 5 show per-participant averages, and the bold curve shows the overall average across participants. Detailed results for each participant are provided in Appendix B.3. After completing the rating task, participants completed the NASA Task Load Index (TLX) survey (Hart & Staveland, 1988), which assesses cognitive workload on a 1 (low) to 7 (high) scale.

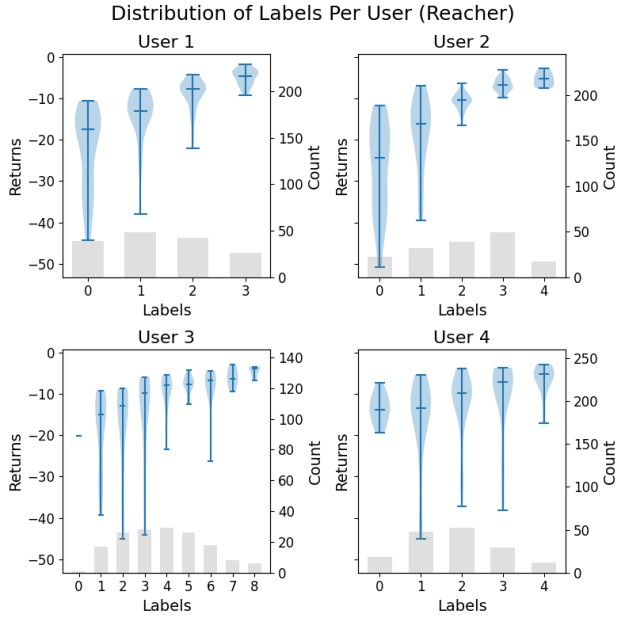

*Figure 4.* The distribution (grey) of rating classes in `Reacher` from four random user study participants, along with the distribution (blue) of environment returns within each rating class.

We identify five findings from the user study. First, user ratings exhibited significant class imbalance, with some participants assigning only a single trajectory to certain rating classes. Second, rating behavior varied substantially across participants, with individuals using different numbers of rating classes ranging from 4 to 12. Third, humans frequently assigned different ratings to trajectories with similar envi-

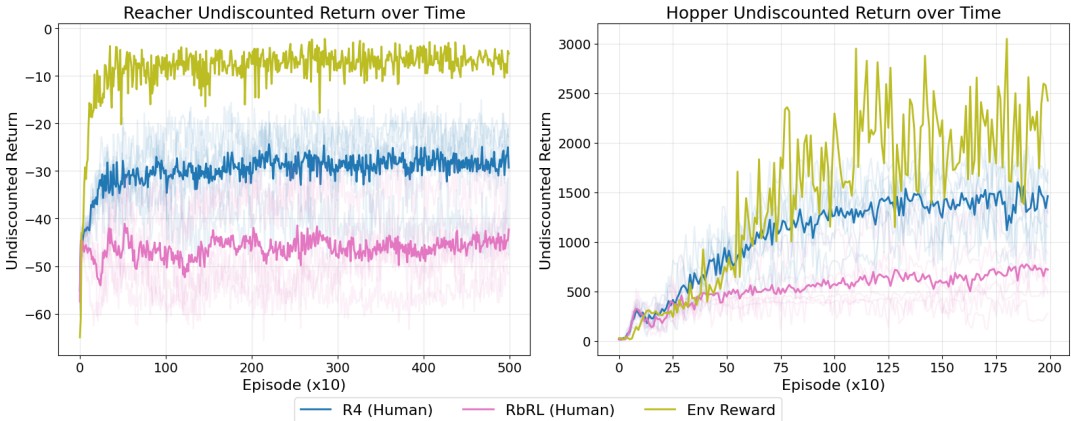

*Figure 5.* SAC performance with R4 and RbRL reward functions learned from trajectory ratings collected in the user study.

ronment returns, unlike simulated ratings, highlighting the value of real human feedback for capturing preferences not aligned with the environment reward (see Figure 4 and Appendix B.3 for the first three findings). Fourth, despite this heterogeneity, R4 achieved significantly faster learning and higher returns in both domains ($p < 0.006$) (see Figure 5 and Appendix B.3, Table 2). Fifth, comparing R4 trained on simulated versus human ratings shows only a small performance gap (see Appendix B.3, Figure 8). This suggests that R4 is robust to the variability of human feedback and that its benefits are not limited to idealized, noise-free settings.

Next, in Table 1, we report the mean NASA TLX scores ($\pm$ standard deviation, STD) from the user study. Participants reported an average overall workload of 2.05 of out 7, where the overall workload was calculated as the average across the survey's subscales. This suggests that providing ratings was generally not perceived as difficult, providing further evidence that ratings are a feasible method of feedback even in continuous-control environments.

### 6.4. Ablations with Simulated Ratings

To assess the impact of the dynamic feedback schedule and stratified trajectory sampling, we ablate these components and evaluate performance in three domains under the online feedback setting. We find that in two of the three domains, R4 performs comparably without these additions. Using either technique in isolation is sufficient to sustain performance, while combining both yields small but consistent improvements (see Appendix C, Figure 22). Next, we evaluate reward quality and report the Trajectory Alignment Coefficient (TAC) (Muslimani et al., 2025) in Appendix C, Table 4. TAC measures how similarly two reward functions rank a set of trajectories. We compute TAC between each method's learned reward and the environment reward (where the simulated ratings were generated from). Results show that R4 produces reward functions that are generally

better aligned with the environment reward compared to the other baselines. Finally, we examine how the number of rating bins affects R4 and RbRL in Appendix C, Figure 25, with R4 remaining stable while RbRL degrades.

*Table 1.* Mean NASA TLX scores ($\pm$ STD) from the user study.

| Metric | Mean Score | Lower is better? |
|---|---|---|
| Overall Workload | $2.05 \pm 0.53$ | ✓ |
| Mental Demand | $2.29 \pm 1.38$ | ✓ |
| Physical Demand | $1.57 \pm 0.98$ | ✓ |
| Hurried | $1.71 \pm 0.76$ | ✓ |
| Effort | $2.29 \pm 0.76$ | ✓ |
| Insecure | $1.29 \pm 0.49$ | ✓ |
| Success | $4.86 \pm 0.69$ | ✗ |

## 7. Conclusion

Reward design remains a fundamental challenge in RL. Therefore, we propose R4, a theoretically grounded algorithm for learning reward functions from multi-class human ratings. Unlike prior work, R4 treats ratings as ordinal feedback and optimizes a rank-based mean squared error loss, allowing the reward model to better exploit the rating structure in the labeled trajectories. To demonstrate the utility of R4, we first provide a theoretical analysis showing that it yields minimal and complete solutions under mild assumptions. Next, we empirically demonstrate its effectiveness across both offline and online feedback scenarios. Lastly, we conduct a human-user study in which participants provide ratings to robotic tasks. Despite high variability between participants, R4 consistently outperforms RbRL. Participants also report low cognitive workload when providing ratings, supporting the practical utility of rating-based reward learning. Overall, our results represent an important step toward reward learning methods that maintain theoretical rigor while effectively leveraging real human feedback.

## Acknowledgements

Part of this work has taken place in the Intelligent Robot Learning Lab at the University of Alberta, which is supported in part by research grants from Alberta Innovates; Alberta Machine Intelligence Institute (Amii); a Canada CI-FAR AI Chair, Amii; Digital Research Alliance of Canada; Mitacs; and the National Science and Engineering Research Council.

## Impact Statement

In this work, we propose a new algorithm for learning from human feedback. If such a system were deployed in a real-world setting, it is important that the human feedback comes from a diverse group of participants to ensure the system does not become biased toward any particular group. Additionally, we conducted a human-user study to evaluate the efficacy of our algorithm with real human feedback. Our study was approved by the appropriate external ethics committee, and all participants provided informed consent prior to participation.

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

# Appendix

## A. Theory Results

### A.1. Derivative of RbRL Loss function

The RbRL (White et al., 2024) loss function is defined as:

$$\mathcal{L}_{\text{RbRL}} = \mathbb{E}_\tau \left( -\sum_{i=0}^{n-1} \mu_i \log(Q(\tau_i)) \right) \tag{6}$$

Where $\mu_i$ is the indicator function for the assigned label, ie. $\mu_i = 1$ when the trajectory $\tau_i$ is assigned the label class $i$ in the dataset, and 0 otherwise. Furthermore, the function $Q$ is defined as:

$$Q(\tau_i) = \frac{e^{-k(\hat{G}_i - B_i)(\hat{G}_i - B_{i+1})}}{\sum_j e^{-k(\hat{G}_j - B_j)(\hat{G}_j - B_{j+1})}} \tag{7}$$

Here, $\{B_i\}_{i=0}^{N-1}$ are class decision boundaries and $k$ is a hyperparameter. We write $\hat{G}_i$ instead of $\hat{G}_\theta(\tau_i)$ for convenience to denote the the normalized predicted return. The derivative of this loss is:

$$\frac{\partial \mathcal{L}_{\text{RbRL}}}{\partial \hat{G}_i} = \mathbb{E}_\tau \left( k \sum_j (\mu_j - Q(\tau_i))(2\hat{G}_i - B_j - B_{j+1}) \right) \tag{8}$$

This result is not new and is presented only because it is used in the proof of proposition 1.

### A.2. Proofs

#### A.2.1. PROOF FOR PROPOSITION 1

**Part 1:** Since $r^*$ is the deterministic data generating reward function, $c(\tau_i) < c(\tau_j) \implies G^*(\tau_i) < G^*(\tau_j)$, where $G^*$ is the trajectory return using the reward function $r^*$. Furthermore, If we try to rank the trajectories according to their corresponding $G^*$, we will recover their $c(\tau_i)$. Hence, $\hat{R}_\theta(\tau_i) = c(\tau_i)$ for all $i$ when $\hat{r}_\theta = r^*$. This implies that $\mathcal{L}_{\text{rMSE}}(\theta) = 0$ when $\hat{r}_\theta = r^*$. Hence $r^* \in \arg\min_\theta \mathcal{L}_{\text{rMSE}}(\theta)$.

**Part 2:** To show that $r^* \notin \arg\min_\theta \mathcal{L}_{\text{RbRL}}(\theta)$ in general, providing a counterexample suffices. Equation 8 shows that $\mathcal{L}_{\text{RbRL}}$ is minimized when the return for each trajectory in a rating class is exactly equal to either $B_i$, $B_{i+1}$, or $\frac{B_i + B_{i+1}}{2}$ (White et al., 2024). Consider the reward function $r^*$ to assign the return of $\frac{B_i + B_{i+1}}{2} + \epsilon$ for some $0 < \epsilon < \frac{B_{i+1} - B_i}{2}$ to each trajectory in the same rating class. Such $r^*$ would not belong to the solution class of $\mathcal{L}_{\text{RbRL}}$. Hence, $r^* \notin \arg\min_\theta \mathcal{L}_{\text{RbRL}}(\theta)$ in general. □

#### A.2.2. PROOF OF THEOREM 1

First, let us assume that there exists some reward function $r_\theta \in \mathcal{R}$.

Since $r_\theta \in \mathcal{R}$,

$$c(\tau_i) < c(\tau_j) \implies G_\theta(\tau_i) < G_\theta(\tau_j)$$

Therefore, if we pick one trajectory from each rating class (without loss of generality):

$$c(\tau_0) < c(\tau_1) < \cdots < c(\tau_{n-1}), \quad \text{where } c(\tau) \in \{0, 1, \cdots, n-1\}$$
$$\implies G_\theta(\tau_0) < G_\theta(\tau_1) < \cdots < G_\theta(\tau_{n-1})$$

This implies that the predicted ranks $\{\hat{R}(\tau_i)\}_{i=0}^{n-1}$ of the $\{G_\theta(\tau_i)\}_{i=0}^{n-1}$ will also follow the same order:

$$\hat{R}(\tau_0) < \hat{R}(\tau_1) < \cdots < \hat{R}(\tau_{n-1}), \quad \text{where } \hat{R} \in \{0, 1, \cdots, n-1\} \tag{9}$$

$$\implies \frac{1}{n} \sum_{i=0}^{n-1} \left( c(\tau_i) - \hat{R}(\tau_i) \right)^2 = 0, \quad \text{Both } c \text{ and } \hat{R} \text{ are distinct integers in [0, n-1] (3)} \tag{10}$$

$$\implies r_\theta \in \arg\min_\theta \mathcal{L}_{\text{rMSE}}(\theta) \tag{11}$$

Therefore, $r_\theta \in \mathcal{R} \implies r_\theta \in \arg\min_\theta \mathcal{L}_{\text{rMSE}}(\theta)$.

Now, let us assume that there exists a reward function $r_\theta \in \arg\min_\theta \mathcal{L}_{\text{rMSE}}(\theta)$.

Let us pick one trajectory from each bin (without loss of generality):

$$\implies c(\tau_0) < c(\tau_1) < \cdots < c(\tau_{n-1})$$

Now, since $r_\theta \in \arg\min_\theta \mathcal{L}_{\text{rMSE}}(\theta)$,

$$\frac{1}{n} \sum_{i=0}^{n-1} \left( c(\tau_i) - \hat{R}(\tau_i) \right)^2 = 0$$

$$\implies c(\tau_i) = \hat{R}(\tau_i), \forall i$$

$$\implies \hat{R}(\tau_0) < \hat{R}(\tau_1) < \cdots < \hat{R}(\tau_{n-1})$$

$$\implies G_\theta(\tau_0) < G_\theta(\tau_1) < \cdots < G_\theta(\tau_{n-1})$$

Therefore, $c(\tau_i) < c(\tau_j) \implies G_\theta(\tau_i) < G_\theta(\tau_j)$, which implies that $r_\theta \in \mathcal{R}$. Hence, using both of these results, $r_\theta \in \mathcal{R} \iff r_\theta \in \arg\min_\theta \mathcal{L}_{\text{rMSE}}(\theta)$ □

### A.3. Relaxing Assumption 3

*Proof of Theorem 2.*
The proof follows a similar structure as the proof for Theorem 1.

First, suppose that there exists some reward function $r_\theta \in \mathcal{R}$.

Since $r_\theta \in \mathcal{R}$, if we pick one trajectory from each rating class (without loss of generality):

$$c(\tau_0) < c(\tau_1) < \cdots < c(\tau_{n-1}), \quad \text{where } c(\tau) \in \{0, 1, \cdots, n-1\}$$
$$\implies G_\theta(\tau_0) < G_\theta(\tau_1) < \cdots < G_\theta(\tau_{n-1})$$

Since $\{G_\theta(\tau_i)\}_{i=0}^{n-1}$ follow the same order as $\{c(\tau_i)\}_{i=0}^{n-1}$, and the predicted ranks $\{\hat{R}(\tau_i)\}_{i=0}^{n-1}$ differs from the true ranks by at most $\epsilon$,

$$\implies \frac{1}{n} \sum_{i=0}^{n-1} \left( \hat{R}(\tau_i) - c(\tau_i) \right)^2 \leq \epsilon^2$$

$$\implies \mathcal{L}_{\text{rMSE}}(\theta) \leq \epsilon^2$$

$$\implies r_\theta \in \mathcal{R}_{\text{rMSE}}$$

Now, let us assume that there exists a reward function $r_\theta \in \mathcal{R}_{\text{rMSE}}$:

If we pick one trajectory from each rating class (without the loss of generality), then:

$$c(\tau_0) < c(\tau_1) < \cdots < c(\tau_{n-1})$$

Since $r_\theta \in \mathcal{R}_{\text{rMSE}}$,

$$\implies \mathcal{L}_{\text{rMSE}}(\theta) \leq \epsilon^2 \tag{12}$$

$$\implies \frac{1}{n} \sum_{i=0}^{n-1} \left( \hat{R}(\tau_i) - c(\tau_i) \right)^2 \leq \epsilon^2 \tag{13}$$

To conclude the proof, we must show that the predicted returns $\{G_\theta(\tau_i)\}_{i=0}^{n-1}$ preserve the same ordering as the class labels $\{c(\tau_i)\}_{i=0}^{n-1}$. If the ordering is preserved, then $r_\theta \in \mathcal{R}$ immediately follows.

Consider the cases where the ordering is violated because of the reward function[4]. For $\epsilon < 0.5$, the "least harmful" violation (i.e., the one that produces the smallest possible $\mathcal{L}_{\text{rMSE}}$ while still breaking the ordering because of the returns from the reward function) occurs when exactly two adjacent elements swap their order, while all other predictions are correct. Without loss of generality, suppose these elements are at indices $k$ and $k+1$, and that $\hat{R}(\tau_i) = c(\tau_i)$ for all $i \notin \{k, k+1\}$.

For example, if the true class labels are $[0, 1, 2]$, then the smallest-error misordering happens when the predictions are $[1 - \epsilon, 0 + \epsilon, 2]$: the first two items are swapped but deviate from their true labels by only $\epsilon$.

This case gives the minimum possible $\mathcal{L}_{\text{rMSE}}$ under an incorrect ordering, which equals $\frac{2(1-\epsilon)^2}{n}$, determined as follows:

$$\frac{1}{n} \sum_{i=0}^{n-1} \left( \hat{R}(\tau_i) - c(\tau_i) \right)^2 = \frac{1}{n} \left[ \underbrace{\sum_{i=0}^{k-1} \left( \hat{R}(\tau_i) - c(\tau_i) \right)^2}_{=0} + (\hat{R}(\tau_k) - c(\tau_k))^2 \right.$$

$$\left. + (\hat{R}(\tau_{k+1}) - c(\tau_{k+1}))^2 + \underbrace{\sum_{i=k+2}^{n-1} \left( \hat{R}(\tau_i) - c(\tau_i) \right)^2}_{=0} \right]$$

$$= \frac{(\hat{R}(\tau_k) - c(\tau_k))^2 + (\hat{R}(\tau_{k+1}) - c(\tau_{k+1}))^2}{n}$$

$$= \frac{(1 - \epsilon - 0)^2 + (\epsilon - 1)^2}{n}$$

$$= \frac{2(1 - \epsilon)^2}{n}. \tag{14}$$

Therefore, to exclude such incorrect orderings from the relaxed solution set $\mathcal{R}_{\text{rMSE}}$, we require that the lower bound on the error of an invalid solution exceed $\epsilon^2$. More specifically:

$$\frac{2(1 - \epsilon)^2}{n} > \epsilon^2$$

$$\implies 0 \leq \epsilon < \frac{\sqrt{2n} - 2}{n - 2}$$

which is the bound on $\epsilon$ in assumption 4. Now, since $\epsilon$ satisfies this bound, continuing from 13, we can be sure that:

$$G_\theta(\tau_0) < G_\theta(\tau_1) < \cdots < G_\theta(\tau_{n-1})$$
$$\implies r_\theta \in \mathcal{R}$$

Combining these two results, we have shown that under assumptions 1, 2, and 4:

$$r_\theta \in \mathcal{R} \iff r_\theta \in \mathcal{R}_{\text{rMSE}}$$

---

[4]Since we only want to exclude the reward functions where the returns from the reward function do not follow the ordering, we consider the cases with $\epsilon < 0.5$. Otherwise the ranking function loses its meaning and we start misordering things because of the ranking function

□

## A.4. Support for Fast-Soft Rank

Here, we present a few outputs from the fast-soft ranking algorithm, to justify Assumption 3. With an appropriate value of `regularization_strength`, the ranking operator outputs the true ranks of all elements in most cases.

*Listing 1.* Sample Outputs From Soft Rank

```
for _ in range(10):
    x = np.random.uniform(0,100,10)
    print(soft_rank(x, regularization_strength=0.065))

'''
Outputs:
[10.   7.   5.   4.   6.   9.   1.   3.   8.   2.]
[ 7.   9.   6.   3.   1.  10.   4.   2.   8.   5.]
[ 5.   3.   4.   6.   7.   8.   2.   9.   1.  10.]
[ 6.   3.   2.   8.   5.   4.   1.  10.   9.   7.]
[ 6.   1.   8.   9.   4.   7.   3.   2.   5.  10.]
[ 8.   3.   2.   4.   5.   9.  10.   6.   1.   7.]
[ 3.   6.   5.   1.   7.   2.   4.  10.   9.   8.]
[ 7.   1.   6.   5.   3.   8.   2.   4.   9.  10.]
[ 1.   3.   8.   9.   6.   5.   2.   4.   7.  10.]
[ 7.   6.   9.   8.   5.   4.   2.  10.   1.   3.]
'''
```

Moreover, we study the impact of the fast-soft-ranking (Blondel et al., 2020) regularization strength in R4 for the `Inverted Double Pendulum` environment. In Figure 6, we plot the undiscounted return averaged over the last 100 episodes as a function of the regularization strength. The plot shows that 83% of the runs with regularization strengths between 0.065 and 1 learn a successful policy.

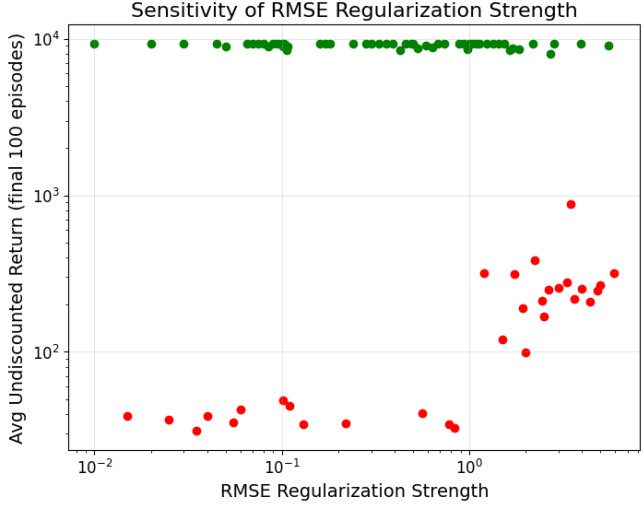

*Figure 6.* Regularization strength of fast-soft ranking (Blondel et al., 2020) vs final learned policy return for `Inverted Double Pendulum`.

## A.5. Analysis of rMSE Objective Under Label Noise

In this section, we analyze the behavior of the ranking mean squared error (rMSE) objective under noisy rating labels. The analysis complements the main theoretical results, which assume noise-free ratings, by characterizing the solution learned

by rMSE when labels are corrupted.

### A.5.1. SETUP

Here, we consider the same setting as Section A.2.2, but relax Assumption 1 by allowing noisy ratings. Consider $c^*(\tau) \in \{0, 1, \cdots, n-1\}$ be the noise-free (true) label of trajectory using the true reward function $r^*$. Let $c(\tau)$ represent the noisy rating sampled from a conditional distribution as follows:

$$c(\tau) \sim P(\cdot|c^*(\tau)) \tag{15}$$

The training batch for calculating rMSE loss is constructed by sampling one trajectory from each rating class. When labels are noisy, the sampling procedure induces an additional dependence between the sampled trajectories and the observed labels. However, for the purpose of analyzing the effect of label noise on the objective, it is convenient to separate two sources of randomness. In the following analysis we ignore the randomness due to sampling and focus our analysis on the expected loss over the label noise.

### A.5.2. EXPECTED LOSS UNDER NOISE

We consider the expected loss under label, conditioned on trajectory $\tau$, as follows:

$$\mathbb{E}\left[(\hat{R}(\tau) - c(\tau))^2 \mid \tau\right]. \tag{16}$$

Let $\mu(\tau) = \mathbb{E}\left[c(\tau) \mid \tau\right]$. Therefore, the above equation can be written as:

$$\mathbb{E}\left[(\hat{R}(\tau) - c(\tau))^2 \mid \tau\right] = \mathbb{E}\left[(\hat{R}(\tau) - \mu(\tau) + \mu(\tau) - c(\tau))^2 \mid \tau\right] \tag{17}$$

$$= \mathbb{E}\left[(\hat{R}(\tau) - \mu(\tau))^2 + (\mu(\tau) - c(\tau))^2 - 2(\hat{R}(\tau) - \mu(\tau))(\mu(\tau) - c(\tau)) \mid \tau\right]. \tag{18}$$

Since $(\hat{R}(\tau) - \mu(\tau))^2$ is constant with respect to $c(\tau)$, $\mathbb{E}\left[(\hat{R}(\tau) - \mu(\tau))^2 \mid \tau\right] = (\hat{R}(\tau) - \mu(\tau))^2$. Similarly for other terms, $\mathbb{E}\left[(\mu(\tau) - c(\tau))^2 \mid \tau\right] = \mathrm{Var}(c(\tau))$, and Expectation of the third term goes to $0$. Therefore,

$$\mathbb{E}\left[(\hat{R}(\tau) - c(\tau))^2 \mid \tau\right] = \left(\hat{R}(\tau) - \mu(\tau)\right)^2 + \mathrm{Var}(c(\tau)). \tag{19}$$

Hence, the loss is minimized when $\hat{R} = \mathbb{E}\left[c(\tau) \mid \tau\right]$. This shows that, under label noise, rMSE implicitly fits the conditional expectation of the observed rating. Therefore, following Theorem 2 rMSE learns a correct order of trajectories when $c^*(\tau_i) > c^*(\tau_j) \implies \mu(\tau_i) > \mu(\tau_j)$. This means that even if individual noisy ratings $c(\tau)$ frequently violate the true trajectory ordering, the rMSE objective will still successfully recover the true reward order, provided the noise process (Equation 15) does not systematically invert the expected ratings (e.g., a highly biased distribution where a truly poor trajectory receives a higher average rating than a truly optimal one).

Order corrupting noises such as adversarial ratings, might be detrimental to rMSE's performance.

## B. Human Subject Study

In this section, we describe our human-subject study. The study was approved by the appropriate ethics review board.

### B.1. User Study Structure

We begin by outlining the structure of the study. At the start of the study, participants provided informed consent. They then read a set of instructions describing the objective of the domain in which they would provide ratings. Participants

were also given guidance on what constituted "good" and "bad" behaviors, following a protocol similar to that used in (Christiano et al., 2017) and (Muslimani & Taylor, 2025). The specific instructions are detailed in the following subsection B.2. After reviewing the instructions, participants completed a short practice session to familiarize themselves with the range of behaviors they would observe. Participants were not limited in the number of practice trajectories they could rate. They then proceeded to rate 159 video clips for `Reacher` and 115 video clips for `Hopper`.

To generate the video clips, we trained a SAC agent for 1000 episodes in `Reacher` and for 1500 episodes in `Hopper`, storing the resulting trajectories. We then randomly sampled $N$ trajectories from each set, where $N$ corresponds to the feedback budget. For `Reacher`, every video clip consisted of 50 environment steps, matching the default episode length. For `Hopper`, episodes can last up to 1000 steps but may terminate earlier. We constructed video clips of at most 200 steps; some clips were shorter due to early termination (e.g., poor behaviors ending in failure), while longer clips typically reflected more successful behaviors. We chose a limit of 200 steps because 50-step clips were too short to adequately assess behavior, whereas substantially longer clips tend to be more challenging for reward learning methods.

After completing the rating task, participants filled out the NASA TLX survey to assess the workload and perceived difficulty of providing ratings (Figure 7). This survey was introduced partway through the study; consequently, the first three participants did not complete it. Participants then completed a demographics survey, the results of which are shown in Table 3. In total, the study took approximately one hour to complete.

After data collection, we randomly sampled 100 ratings from the full set of 159 ratings for `Reacher` and from the full set of 115 ratings for `Hopper`. Using each participant's sampled rating data, we trained a reward function. We then trained a SAC agent in the same environment using the learned reward function. We repeated this process five times to account for randomness in both reward-function learning and SAC training.

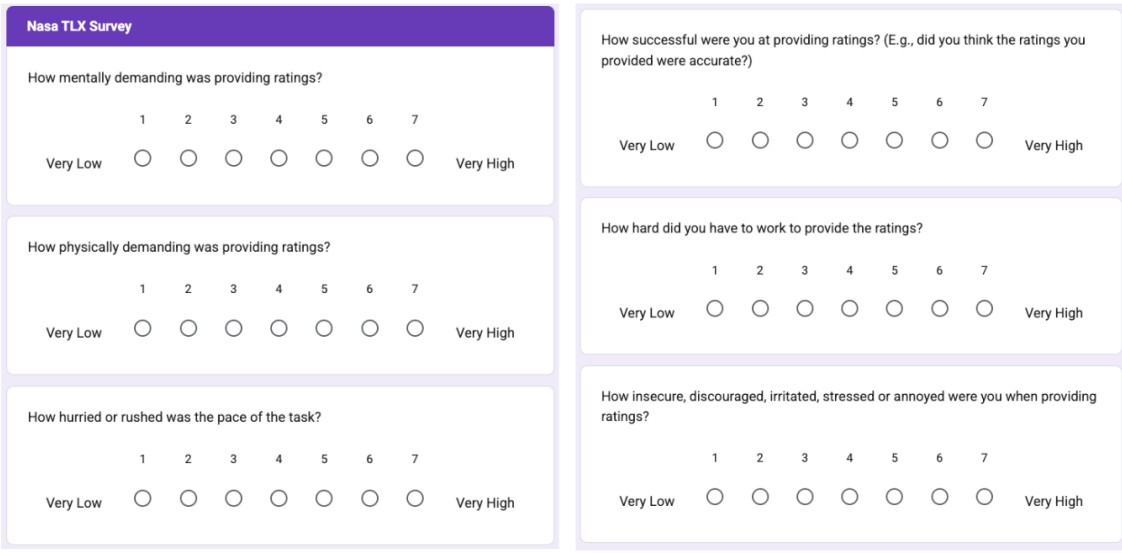

*Figure 7.* NASA TLX Survey provided to participants at the end of the human-subject study.

### B.2. Participant Instructions

Participants were given detailed instructions explaining how to provide ratings, the goal of the domain, and suggestions for interpreting behaviors in the environment.

**Rating Procedure.** Participants were informed that they would view a series of short video clips depicting robot behavior and assign each clip to a discrete rating category. They were free to choose the number of rating classes they preferred, with a minimum of two classes. For example, participants could use a binary scale (e.g., $0-1$) or a more fine-grained scale (e.g., $0-9$), depending on what they found most intuitive. Multiple video clips could be assigned the same rating.

Participants were told that the rating scale was flexible: if a clip had previously been assigned the lowest rating and a worse

clip was later encountered, they could assign a lower value (e.g., $-1$). Participants could replay video clips as many times as needed. They were also allowed to skip a clip without providing a rating if they were uncertain, and were encouraged to do so if they lacked confidence in their assessment.

**Environment Description for `Reacher`.**  Participants were instructed that the objective of the task was for the robot to move its end effector to a red target representing the goal state and to remain at the goal once it was reached. They were informed that actions in this environment corresponded to the amount of torque applied to each joint.

**Behavioral Guidance for `Reacher`.**  Participants were provided with high-level guidance to aid their evaluations. Slower and more controlled movements were described as preferable to faster, unstable motions. Behaviors in which the robot's end effector remained close to the goal for a longer duration were emphasized as higher quality. As a general ordering guideline, participants were told that maintaining proximity to the goal throughout the clip was preferable to reaching the goal multiple times, which in turn was preferable to reaching the goal once and then leaving, with the lowest-quality behaviors failing to approach the goal or exhibiting excessive speed.

**Environment Description for `Hopper`.**  Participants were told that the task involved a one-legged robot attempting to move forward by hopping without losing balance. They were informed that the agent controls the torques applied at each joint. The goal is for the robot to continuously hop forward.

**Behavioral Guidance for `Hopper`.**  To support their evaluations, participants were given qualitative guidance about typical behaviors observed in this environment. Behaviors in which the robot immediately fell over were described as lowest quality, followed by trajectories where the robot briefly attempted to stand before falling. Trajectories that achieved a single hop were considered higher quality, while those exhibiting multiple consecutive hops without falling were described as the highest-quality behaviors.

### B.3. Additional User Study Results

Table 3 shows the demographics along with the expertise of the 8 participants in the human-subject study. Figures 8 compare R4 trained using simulated ratings with R4 trained using human-provided ratings. As expected, simulated ratings yield slightly better performance, likely due to the absence of human noise. However, the performance gap is small, suggesting that R4 remains effective when trained on real, noisy human feedback.

In Figures 9–15 and Figures 16–21, we present individual results from participants providing ratings in the `Reacher` and `Hopper` domains, respectively. For each participant, the left plot shows SAC learning curves using that participant's learned reward function, where lighter lines denote different random seeds and the darker line shows the mean performance.

The middle plot shows the distribution of rating classes provided by each participant. These distributions are highly non-uniform and vary substantially across individuals, with different individuals using different numbers of rating classes.

The right plot shows the distribution of undiscounted environment returns associated with each rating label in the dataset, highlighting differences between human and simulated feedback. Whereas a simulated teacher would give similar ratings to trajectories with similar returns, human participants frequently provide different ratings for trajectories with comparable environment returns.

In Table 2 we present the $t$-tests results comparing the area under the curves and final return between R4 and RbRL for the results presented in Figure 5.

| Environment | Metric | R4 (Mean $\pm$ STD) | RbRL (Mean $\pm$ STD) | t-stat | p-value |
|---|---|---|---|---|---|
| Reacher | Return | $-27.65 \pm 6.71$ | $-44.62 \pm 5.62$ | 3.88 | 0.0047 |
| | AUC | $-14841.43 \pm 2835.59$ | $-23084.79 \pm 3414.72$ | 3.71 | 0.0059 |
| Hopper | Return | $1410.10 \pm 238.75$ | $723.87 \pm 364.44$ | 3.52 | 0.0055 |
| | AUC | $213165.63 \pm 33445.86$ | $105723.26 \pm 42149.56$ | 4.46 | 0.0012 |

*Table 2.* Comparison of final return and AUC between R4 and RbRL using human ratings, with significance assessed via Welch's $t$-test.

*Table 3.* Participant Demographics and Expertise from the user study.

| Question / Category | Count |
| --- | --- |
| **Age** | |
| $18 - 24$ | 2 |
| $25 - 30$ | 5 |
| $60 - 70$ | 1 |
| **Highest Level of Education** | |
| Bachelor's degree | 4 |
| Master's degree | 4 |
| **Gender** | |
| Female | 4 |
| Male | 4 |
| **Race / Ethnicity** | |
| East Asian | 2 |
| South Asian | 3 |
| Middle Eastern or North African | 1 |
| White / Caucasian | 1 |
| Not listed | 1 |
| **Expertise** | |
| Machine Learning (ML) | 1 |
| Other topics in Computing Science (CS) | 1 |
| Reinforcement Learning | 3 |
| Artificial Intelligence (AI) | 2 |
| Not listed above | 1 |

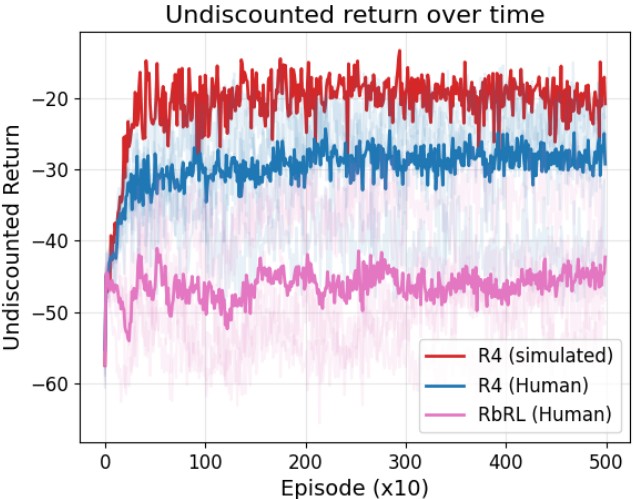

*Figure 8.* Aggregated `Reacher` performance of policies trained on reward models learned from human ratings. Light curves show the average performance across five SAC training runs for each user; the dark curve shows the overall mean across users.

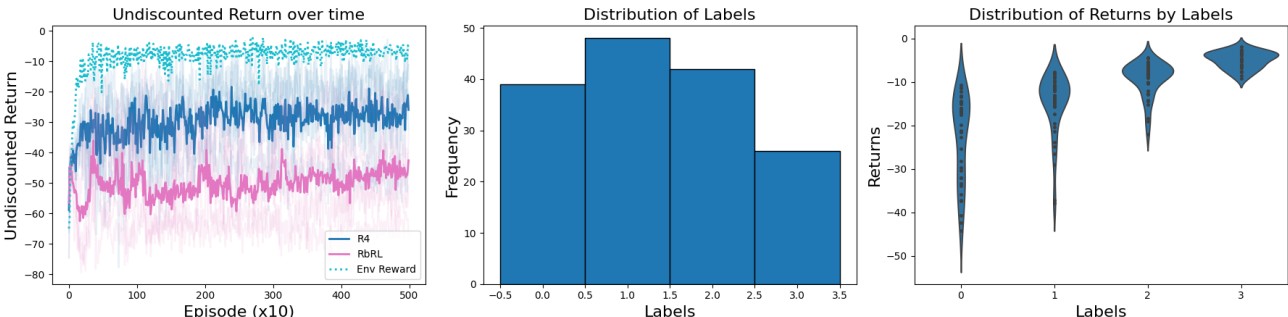

*Figure 9.* `Reacher` data from Participant 1.

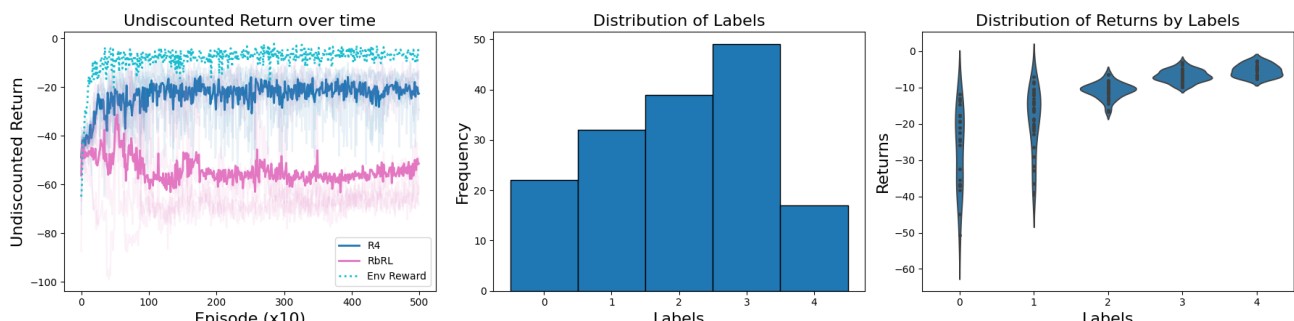

*Figure 10.* `Reacher` data from Participant 2.

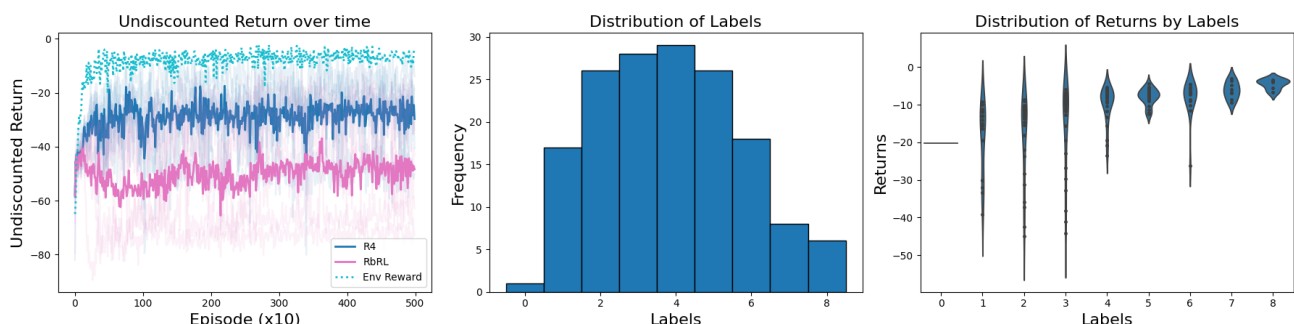

*Figure 11.* `Reacher` data from Participant 3.

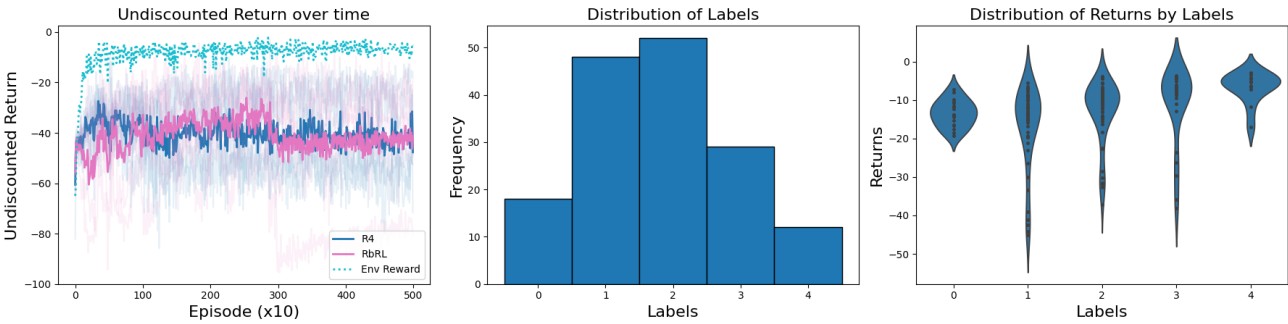

*Figure 12.* `Reacher` data from Participant 4.

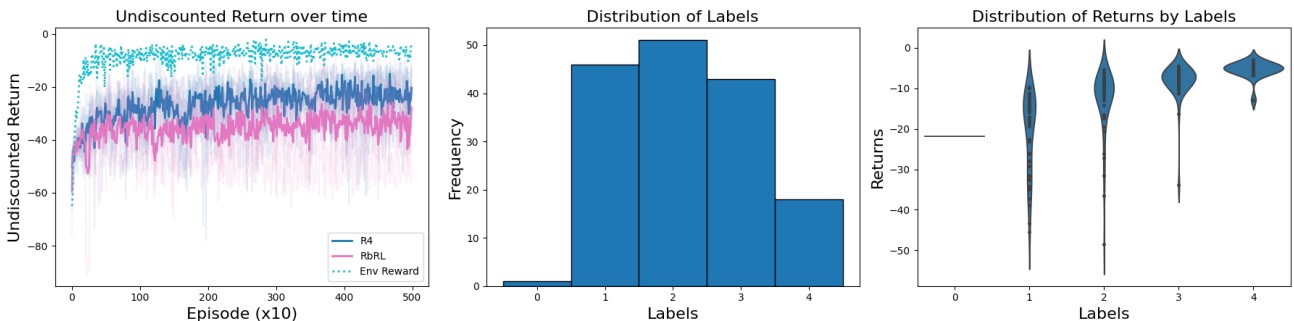

*Figure 13.* `Reacher` data from Participant 5.

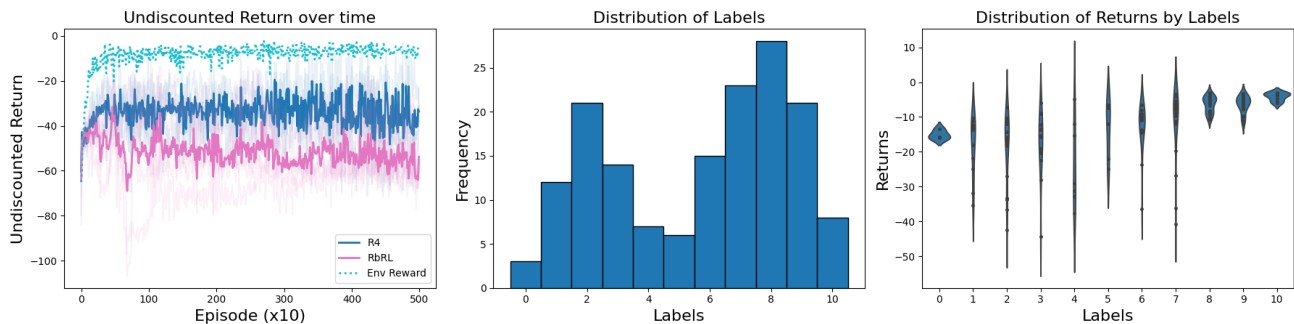

*Figure 14.* `Reacher` data from Participant 6.

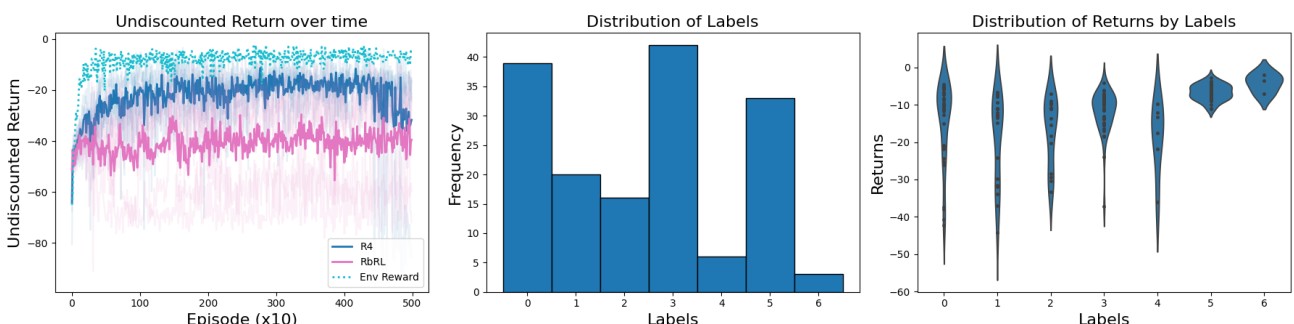

*Figure 15.* `Reacher` data from Participant 7.

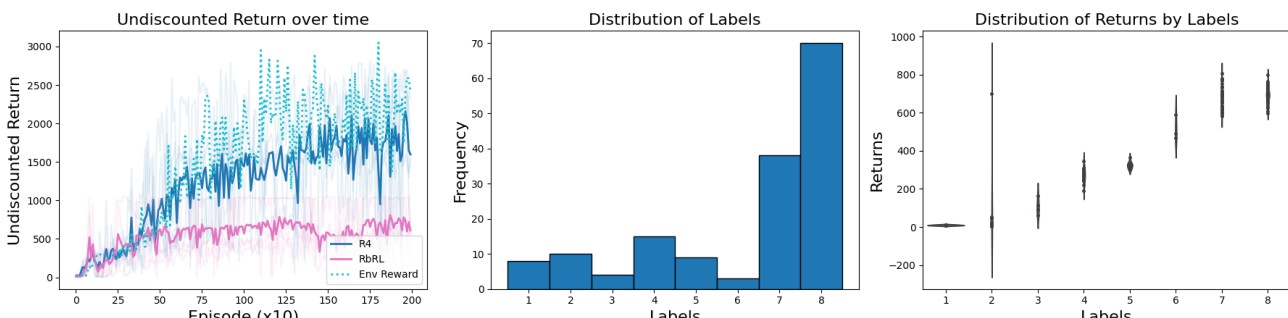

*Figure 16.* `Hopper` data from Participant 1.

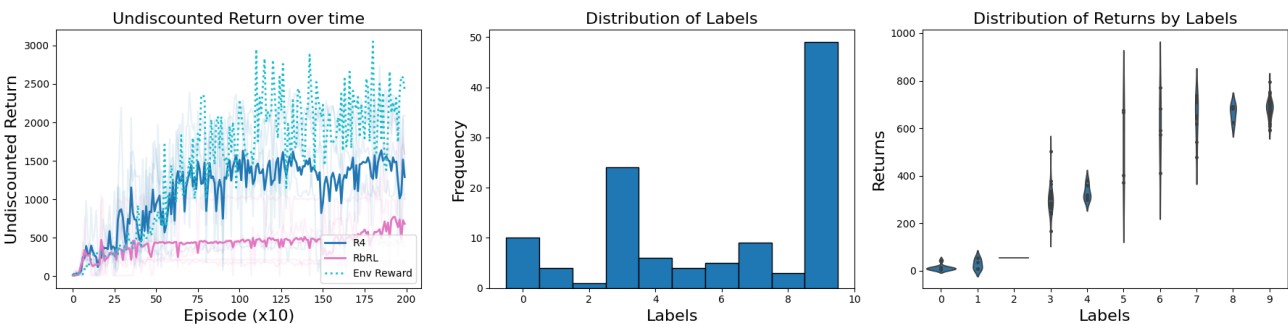

*Figure 17.* `Hopper` data from Participant 2.

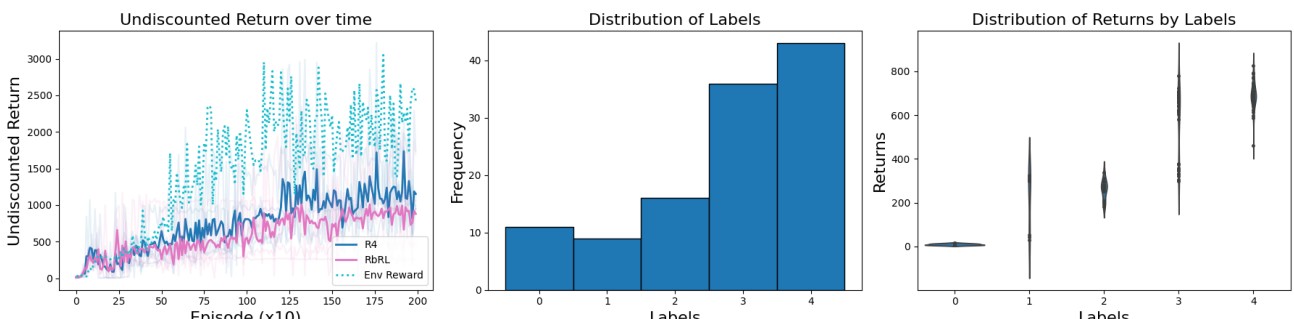

*Figure 18.* `Hopper` data from Participant 3.

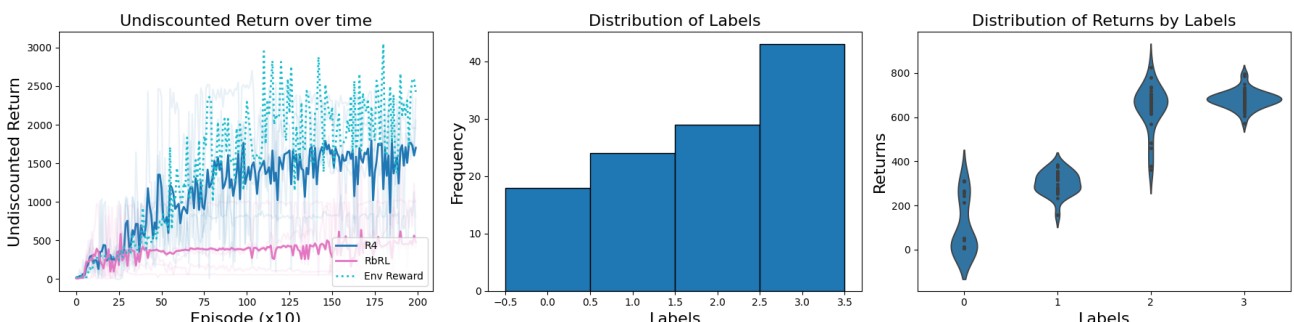

*Figure 19.* `Hopper` data from Participant 4.

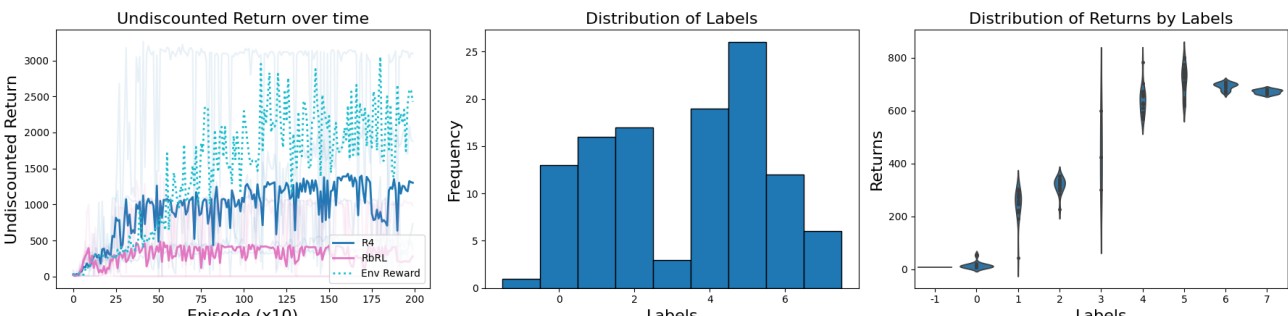

*Figure 20.* `Hopper` data from Participant 5.

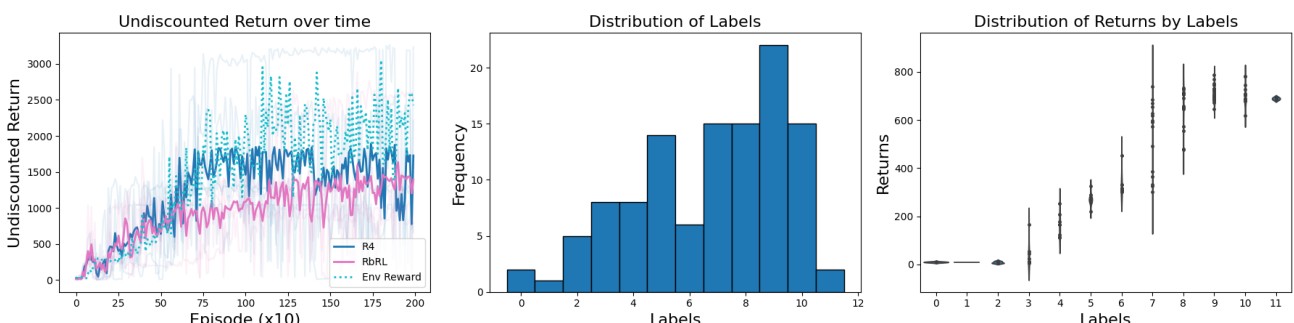

*Figure 21.* `Hopper` data from Participant 6.

## B.4. Dataset Overview

We release the dataset of human ratings collected in the `Reacher` and `Hopper` environments to aid future research on reward learning from human rated feedback, particularly in settings involving noisy, subjective, and heterogeneous human supervision. Apart from the details already provided in previous sections, this section aims to describe the dataset structure and usage:

**Rating Scale and Heterogeneity:** Ratings are absolute, discrete evaluations assigned independently by each annotator. As described in Section B.2, users were allowed to (1) choose their own rating scale, (2) extend the scale if they encountered behavior outside their chosen range, and (3) skip trajectories when uncertain. As a result, rating scales are annotator-specific and not globally normalized. The released dataset preserves all raw ratings without aggregation or rescaling. This design allows researchers to explicitly account for inter-rater variability, inconsistency, and noise, and to develop algorithms better suited to handling these challenges.

**Dataset Statistics:** For each environment and each rater, the dataset contains (1) a set of trajectories, (2) rating labels corresponding to each trajectory, and (3) the undiscounted environment returns associated with each trajectory. For `Reacher`, users employed an average of 6.22 rating classes, with a standard deviation of 2.30. Similarly, for `Hopper`, users employed an average of 6.57 rating classes, with a standard deviation of 2.26. The environment returns of the collected trajectories also exhibit substantial variability: for `Reacher`, average environment returns are $-13.27 \pm 9.02$, ranging from $-45.03$ to $-2.86$, while for `Hopper`, average environment returns are $500.54 \pm 239.97$, ranging from $5.67$ to $825.51$.

**File Structure:** The ratings for each environment are stored in the corresponding folder named `<Env_name>_Human`. Each folder contains three pickle files per user: `pX_labels.pkl`, `pX_returns.pkl`, and `pX_state_action_pairs.pkl`, where `X` denotes the user ID. The file `pX_state_action_pairs.pkl` contains the trajectories shown to the user, while `pX_labels.pkl` and `pX_returns.pkl` contain the corresponding user-provided ratings and undiscounted environment returns, respectively.

**Intended Use and Limitations:** The dataset is intended for research on reward learning from human provided ratings, and robustness to noisy human supervision. Because ratings are subjective and use annotator-specific scales, direct comparison of raw rating values across annotators may be inappropriate without normalization or modeling assumptions. If using this our provided dataset, researchers should account for this heterogeneity when designing learning algorithms or evaluation procedures.

## C. Additional Simulated Experiments

Figure 22 shows the ablation of the dynamic feedback schedule and the stratified trajectory sampling strategy used in R4.

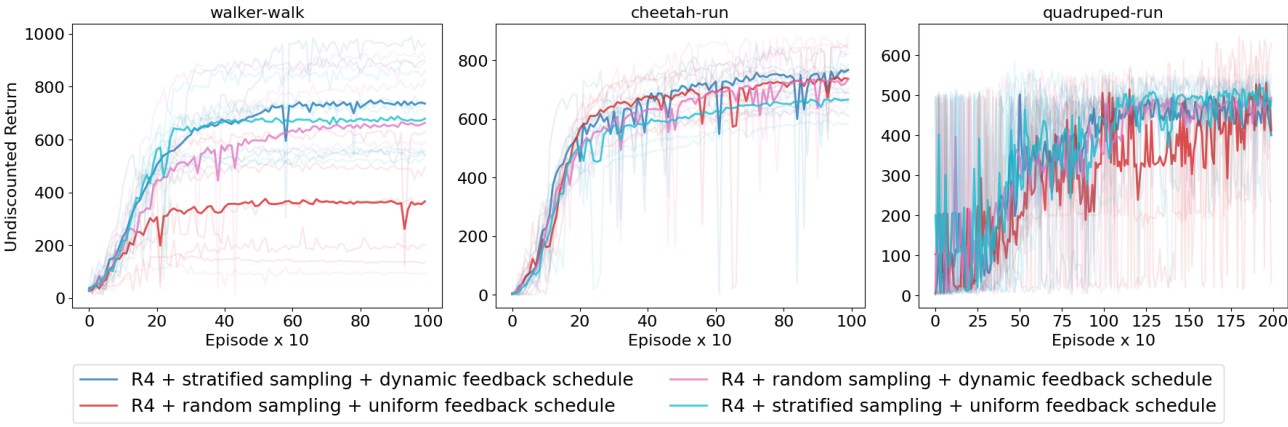

*Figure 22.* We evaluate R4 under ablations of our implementation choices: stratified sampling and the dynamic feedback schedule. We find that both features can improve the base R4 method.

To assess the impact of the dynamic feedback schedule and sampling tricks on the baselines, we tested them with these modifications included. Figure 23 shows that the baselines' performance either remains similar or degrades compared to Figure 3.

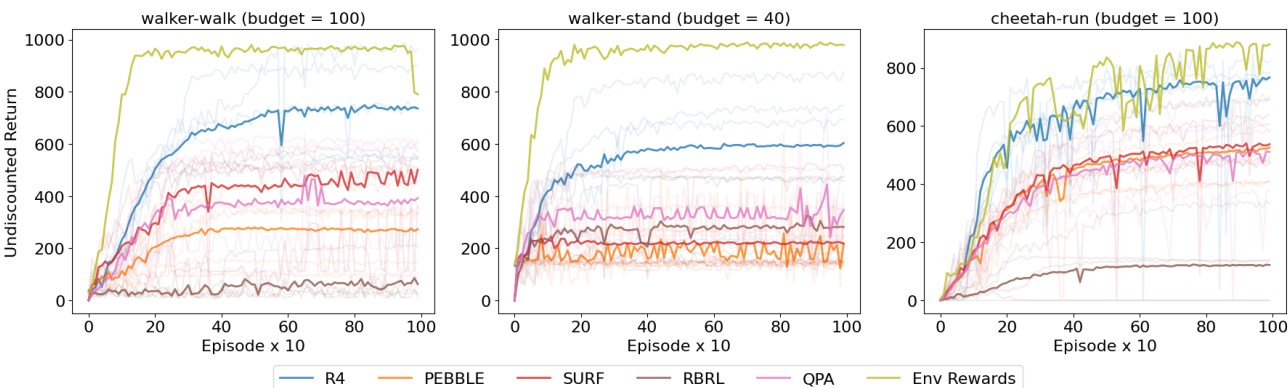

*Figure 23.* Mean undiscounted return (computed using the environment's reward function) versus the number of episodes when training a SAC agent with R4 and various baselines in the online setting. The baselines are allowed to use our dynamic feedback schedule and their respective query sampling tricks.

Second, we evaluate the resilience of R4 to noisy feedback on the `Inverted Double Pendulum` task in the offline setting, where both R4 and RbRL achieve similar final performance under noiseless conditions. We focus on the offline setting because it isolates the effect of noise on reward learning. To simulate noisy human feedback, we randomly select $\eta\%$ of the trajectories in the dataset $\mathcal{D}$ and reassign them to `true_bin`$\pm 1$ with probability $0.5$ each.

Figure 24a shows R4 performance under varying noise levels. While performance naturally decreases as $\eta$ increases, R4 remains robust even at high noise levels. Figure 24b compares R4 with RbRL under the same conditions, showing that RbRL fails even at small noise levels. Notably, R4 with 80% noise achieves performance comparable to RbRL with only 10% noise.

Furthermore, we study the impact of the number of rating classes on R4 in the `Reacher` environment. Figure 25 shows that although RbRL's performance depends significantly on the number of bins, R4 remains consistent.

Overall, these results highlight that R4 is robust to dynamic feedback schedules, resilient to noisy feedback, and largely insensitive to the choice of rating classes, in contrast to RbRL, which is sensitive to all three.

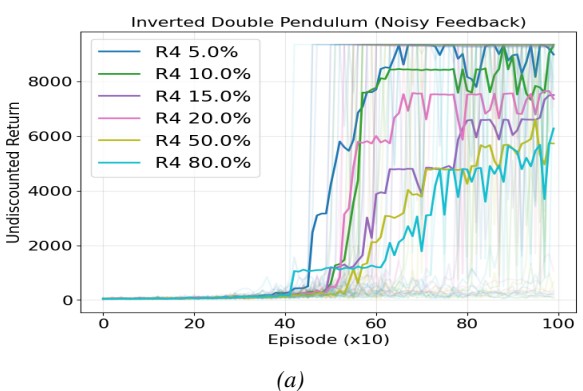 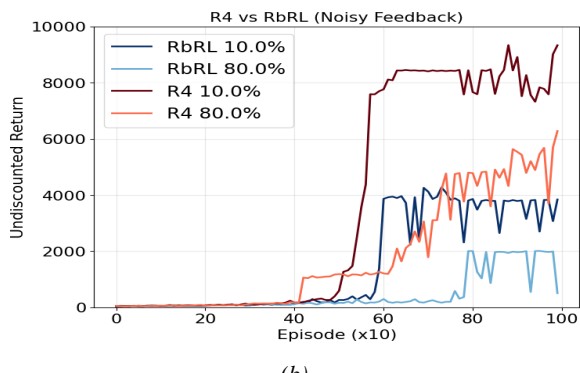

*Figure 24.* (a) R4 objective under varying levels of noise. (b) Comparison of R4 (reds) and RbRL (blues) under different noise levels.

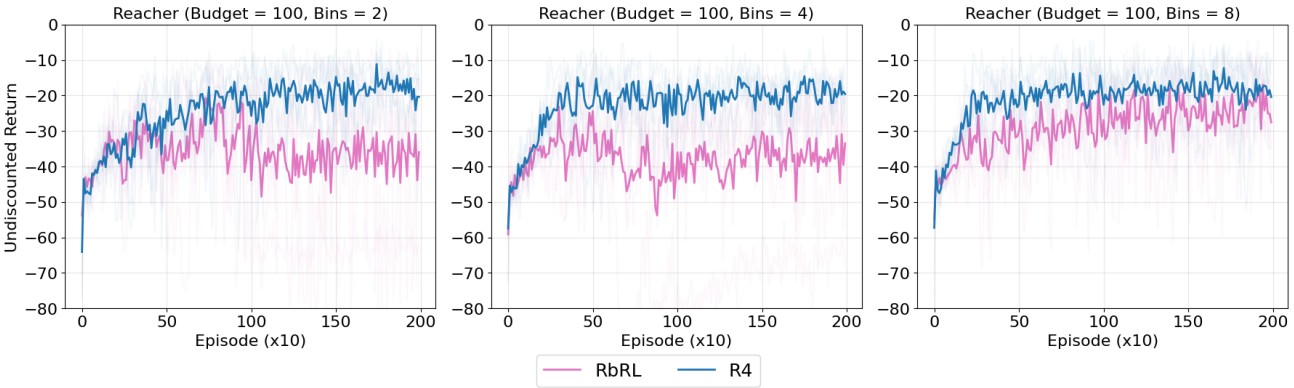

*Figure 25.* Undiscounted return vs number of episodes for reacher with varying number of bins.

## C.1. Quality of Learned Reward Functions

To assess the quality of the reward functions learned by R4 relative to the baselines, we first present a scatter plot comparing undiscounted returns from the learned reward functions against the undiscounted environment returns encountered during a single online run (Figure 26). We show this for three environments and for all methods. The plots indicate that R4 consistently captures a meaningful relationship between learned and actual returns across environments.

Furthermore, to evaluate reward quality quantitatively, we report the Trajectory Alignment Coefficient (TAC) (Muslimani et al., 2025) in Table 4. TAC is a reward alignment metric that measures how similarly two reward functions rank a set of trajectories, where a TAC of 1 indicates perfect alignment and a TAC of $-1$ indicates perfect negative correlation. We compare the reward functions learned by each method with the environment reward functions using TAC. For this comparison, we consider the trajectories encountered during training. Table 4 shows that R4 produces reward functions that are more aligned to the environment reward in two out of three environments.

| Environment | PEBBLE | SURF | RbRL | QPA | R4 |
|---|---|---|---|---|---|
| walker_walk | 0.4681 | 0.5386 | 0.2643 | 0.6521 | 0.7168 |
| cheetah_run | 0.8828 | 0.8366 | 0.0200 | 0.8107 | 0.5956 |
| humanoid_stand | 0.3537 | 0.1640 | $-0.0116$ | 0.1452 | 0.6312 |

*Table 4.* Average TAC scores across environments and methods.

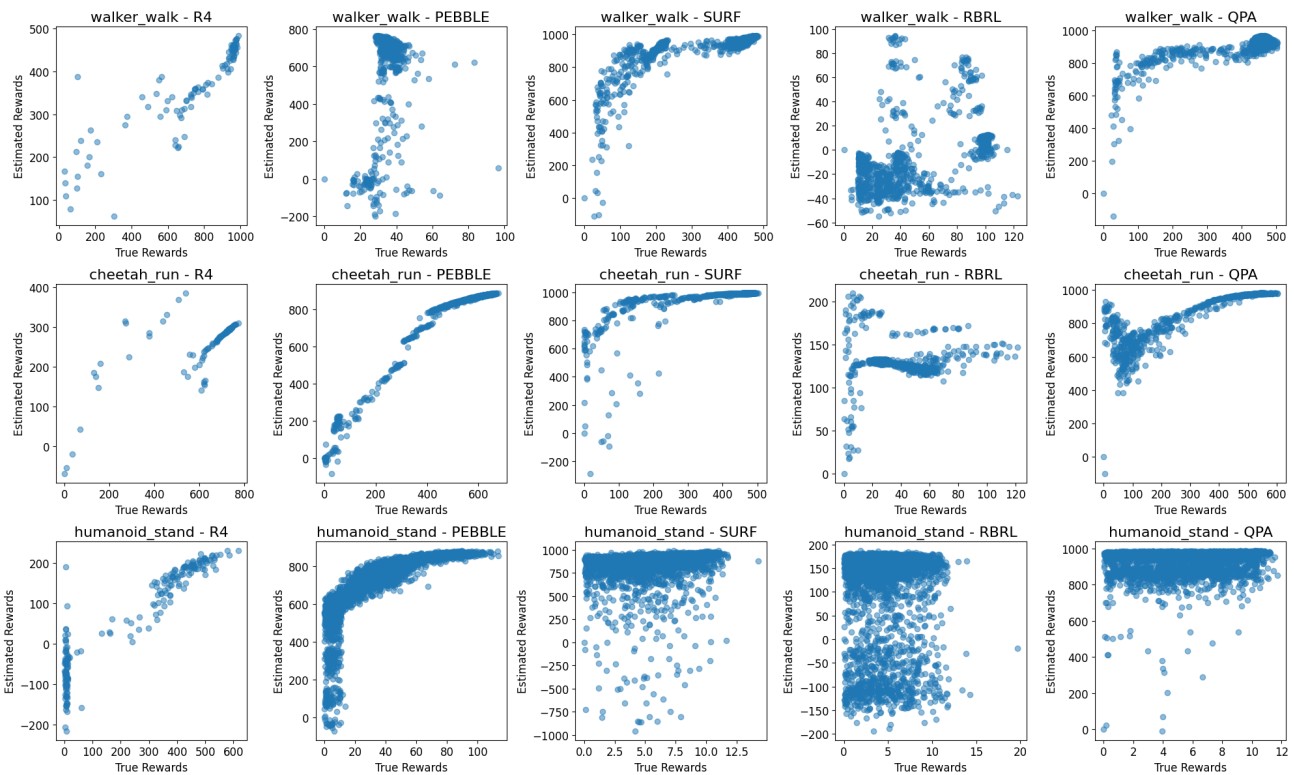

*Figure 26.* Qualitative results

## C.2. Statistical Significance

To assess the statistical significance of our main results presented in Figures 2 and 3, we applied Welch's t-test to two key metrics: the average return over the last 100 episodes and the area under the learning curve (AUC). These metrics capture both the ultimate performance and the overall learning dynamics of each method across multiple random seeds. Table 5 reports the results for the offline feedback runs shown in Figure 2, while Tables 6–9 present analogous results for the online feedback setting in Figure 3. Across nearly all environments, our method demonstrates statistically significant improvements over the baselines in both final return and AUC, indicating not only higher ultimate performance but also more efficient learning.

| Environment | Metric | R4 (Mean $\pm$ STD) | RbRL (Mean $\pm$ STD) | t-stat | p-value |
|---|---|---|---|---|---|
| Reacher | Return | $-19.91 \pm 3.45$ | $-39.13 \pm 13.03$ | 2.85 | 0.0398 |
| | AUC | $-10331.49 \pm 1448.88$ | $-18863.86 \pm 6327.05$ | 2.63 | 0.0527 |
| Inverted DP | Return | $9212.41 \pm 272.56$ | $8485.79 \pm 1239.53$ | 1.15 | 0.3108 |
| | AUC | $441942.60 \pm 58656.92$ | $206676.88 \pm 82616.80$ | 4.64 | 0.0022 |
| Half Cheetah | Return | $3638.23 \pm 1115.80$ | $1746.10 \pm 1380.16$ | 2.13 | 0.0671 |
| | AUC | $282021.30 \pm 98623.15$ | $75385.09 \pm 90477.05$ | 3.09 | 0.0151 |

*Table 5.* Offline feedback setting: Comparison of final return and AUC between R4 and Baseline with Welch's t-test results.

| Environment | Metric | R4 (Mean $\pm$ STD) | PEBBLE (Mean $\pm$ STD) | p-value |
|---|---|---|---|---|
| walker-walk | Return | $736.78 \pm 173.37$ | $246.13 \pm 213.81$ | 0.000 |
| | AUC | $59428.38 \pm 11858.29$ | $18832.11 \pm 15324.24$ | 0.003 |
| walker-stand | Return | $594.24 \pm 223.85$ | $154.46 \pm 31.18$ | 0.000 |
| | AUC | $52901.37 \pm 17831.47$ | $16072.85 \pm 1370.81$ | 0.014 |
| cheetah-run | Return | $747.68 \pm 61.25$ | $463.69 \pm 187.02$ | 0.000 |
| | AUC | $59464.45 \pm 3505.70$ | $36415.47 \pm 13841.43$ | 0.027 |
| quadruped-walk | Return | $500.64 \pm 197.69$ | $107.07 \pm 184.75$ | 0.000 |
| | AUC | $76601.70 \pm 28181.97$ | $25782.51 \pm 7008.07$ | 0.021 |
| quadruped-run | Return | $451.30 \pm 85.90$ | $214.81 \pm 234.10$ | 0.000 |
| | AUC | $70808.57 \pm 14600.77$ | $26447.09 \pm 6189.97$ | 0.002 |
| humanoid-stand | Return | $537.04 \pm 157.46$ | $30.30 \pm 32.46$ | 0.000 |
| | AUC | $43349.13 \pm 16132.70$ | $3171.91 \pm 2630.79$ | 0.007 |

*Table 6.* Online feedback setting: Comparison of final return and AUC between R4 and PEBBLE across environments with Welch's t-test p-values.

| Environment | Metric | R4 (Mean $\pm$ STD) | SURF (Mean $\pm$ STD) | p-value |
|---|---|---|---|---|
| walker-walk | Return | $736.78 \pm 173.37$ | $366.89 \pm 205.47$ | 0.000 |
| | AUC | $59428.38 \pm 11858.29$ | $30528.74 \pm 16237.83$ | 0.023 |
| walker-stand | Return | $594.24 \pm 223.85$ | $292.50 \pm 114.15$ | 0.000 |
| | AUC | $52901.37 \pm 17831.47$ | $28487.92 \pm 10485.57$ | 0.053 |
| cheetah-run | Return | $747.68 \pm 61.25$ | $496.27 \pm 105.96$ | 0.000 |
| | AUC | $59464.45 \pm 3505.70$ | $39690.18 \pm 8243.98$ | 0.006 |
| quadruped-walk | Return | $500.64 \pm 197.69$ | $142.08 \pm 193.82$ | 0.000 |
| | AUC | $76601.70 \pm 28181.97$ | $28346.64 \pm 5380.05$ | 0.025 |
| quadruped-run | Return | $451.30 \pm 85.90$ | $119.98 \pm 177.57$ | 0.000 |
| | AUC | $70808.57 \pm 14600.77$ | $23723.68 \pm 3381.69$ | 0.002 |
| humanoid-stand | Return | $537.04 \pm 157.46$ | $4.51 \pm 2.73$ | 0.000 |
| | AUC | $43349.13 \pm 16132.70$ | $1113.39 \pm 67.12$ | 0.006 |

*Table 7.* Online feedback setting: Comparison of final return and AUC between R4 and SURF across environments with Welch's t-test p-values.

| Environment | Metric | R4 (Mean $\pm$ STD) | RBRL (Mean $\pm$ STD) | p-value |
|---|---|---|---|---|
| walker-walk | Return | $736.78 \pm 173.37$ | $77.18 \pm 60.08$ | 0.000 |
| | AUC | $59428.38 \pm 11858.29$ | $6359.18 \pm 3429.20$ | 0.000 |
| walker-stand | Return | $594.24 \pm 223.85$ | $238.70 \pm 132.50$ | 0.000 |
| | AUC | $52901.37 \pm 17831.47$ | $21555.53 \pm 8995.78$ | 0.020 |
| cheetah-run | Return | $747.68 \pm 61.25$ | $111.43 \pm 210.31$ | 0.000 |
| | AUC | $59464.45 \pm 3505.70$ | $9548.40 \pm 16798.53$ | 0.003 |
| quadruped-walk | Return | $500.64 \pm 197.69$ | $437.97 \pm 338.88$ | 0.267 |
| | AUC | $76601.70 \pm 28181.97$ | $51671.65 \pm 24218.54$ | 0.217 |
| quadruped-run | Return | $451.30 \pm 85.90$ | $467.09 \pm 27.19$ | 0.225 |
| | AUC | $70808.57 \pm 14600.77$ | $64727.11 \pm 8174.64$ | 0.493 |
| humanoid-stand | Return | $537.04 \pm 157.46$ | $4.91 \pm 3.09$ | 0.000 |
| | AUC | $43349.13 \pm 16132.70$ | $928.10 \pm 63.57$ | 0.006 |

*Table 8.* Online feedback setting: Comparison of final return and AUC between R4 and RBRL across environments with Welch's t-test p-values.

| Environment | Metric | R4 (Mean $\pm$ STD) | QPA (Mean $\pm$ STD) | p-value |
|---|---|---|---|---|
| `walker-walk` | Return | $736.78 \pm 173.37$ | $627.46 \pm 199.33$ | 0.005 |
| | AUC | $59428.38 \pm 11858.29$ | $54360.80 \pm 17751.89$ | 0.649 |
| `walker-stand` | Return | $594.24 \pm 223.85$ | $214.50 \pm 101.02$ | 0.000 |
| | AUC | $52901.37 \pm 17831.47$ | $21386.02 \pm 8224.46$ | 0.020 |
| `cheetah-run` | Return | $747.68 \pm 61.25$ | $501.78 \pm 112.86$ | 0.000 |
| | AUC | $59464.45 \pm 3505.70$ | $39251.21 \pm 6357.56$ | 0.001 |
| `quadruped-walk` | Return | $500.64 \pm 197.69$ | $58.50 \pm 138.00$ | 0.000 |
| | AUC | $76601.70 \pm 28181.97$ | $12646.85 \pm 5822.67$ | 0.009 |
| `quadruped-run` | Return | $451.30 \pm 85.90$ | $62.94 \pm 119.64$ | 0.000 |
| | AUC | $70808.57 \pm 14600.77$ | $14861.70 \pm 8587.97$ | 0.000 |
| `humanoid-stand` | Return | $537.04 \pm 157.46$ | $6.04 \pm 3.04$ | 0.000 |
| | AUC | $43349.13 \pm 16132.70$ | $1115.85 \pm 131.49$ | 0.006 |

*Table 9.* Online feedback setting: Comparison of final return and AUC between R4 and QPA across environments with Welch's t-test p-values.

## D. Implementation Details

This section includes the details necessary to replicate our results. Code will be released if the paper is accepted.

### D.1. Batch Updates

While computing the loss using a single sampled trajectory per class dataset $\mathcal{D}_k$ provides a valid training signal, it can lead to a biased gradient estimate and hinder learning. To improve stability, we perform the soft ranking procedure $B$ times per update step. In each iteration, we sample one trajectory per class dataset, compute predicted returns using $\hat{r}_\theta$, and apply the differentiable sorting algorithm (Blondel et al., 2020) to obtain soft ranks. The resulting $B$ soft rank vectors (of size $n$) are then stacked to form a stacked soft ranks matrix. Correspondingly, we stack the class labels associated with each sampled trajectory into a ratings matrix. We compute the rMSE loss as the mean squared error between the stacked soft ranks and the stacked ratings.

### D.2. Possible Regularization

#### D.2.1. L2 REGULARIZATION

For training our reward functions, we use an L2 regularization loss (with coefficient $\beta$) defined as:

$$\mathcal{L}_{L2} = \mathbb{E}_{\tau_i}\left[|\hat{r}_\theta(\tau_i)|^2\right]$$

#### D.2.2. OUT OF DISTRIBUTION REGULARIZATION

Even though we do not use OOD regularization in our experiments, offline RL literature (Kumar et al., 2020; Li et al., 2021) tells that it might be a good tool to have when learning from an under-specified dataset. The idea is to penalize high predicted rewards (under $\hat{r}_\theta$) for state-action pairs not present in the dataset, $\mathcal{D}$:

$$\mathcal{L}_{\text{OOD}} = \mathbb{E}_{s,a\sim p}\left[\hat{r}_\theta(s,a)\right] - \mathbb{E}_{s,a\sim\mathcal{D}}\left[\hat{r}_\theta(s,a)\right]$$

Here, $p$ is a distribution used to sample out-of-distribution state-action pairs. The first term in $\mathcal{L}_{\text{OOD}}$ penalizes high predicted reward values for out-of-distribution pairs, while the second term prevents the learned reward function from collapsing to large negative values. Without the second term, the learned reward function could trivially assign large negative values to all the state-action pairs, including those in the dataset.

### D.3. Online Implementation Details

#### D.3.1. STRATIFIED SAMPLING HUERISTIC

In the online feedback setting with a limited budget, it is crucial to ask for feedback on the trajectories that maximally increase the information provided to the reward function. To achieve this, we maintain a dataset of the latest 50 trajectories

and propose the following trajectory sampling heuristic:

1. **Sorting:** We first sort the trajectories according to their predicted returns.

2. **Sampling:** We then sample $1/3$ of the trajectories from the top 30% of this sorted set, and $2/3$ of the trajectories are sampled from the remaining 70%.

3. **Sub-trajectory selection:** For each sampled trajectory, we extract a sub-trajectory of length $\delta$. This sub-trajectory is either chosen (1) uniformly at random, or (2) as the sub-trajectory with the highest predicted return. Each of these option is applied with equal probability (0.5).

Such a querying mechanism ensures that the queries capture both the typical behavior of the agent and highly informative segments.

### D.3.2. DYNAMIC FEEDBACK SCHEDULE

We apply a dynamic feedback schedule, collecting feedback more frequently at the beginning of training to provide an initial bias to the reward function. Early feedback helps ground the reward model in the environment and mitigates the impact of random neural network initialization on the agent's learning. Then, as training progresses, we gradually reduce the feedback frequency. This ensures that the reward model is updated with more informative trajectory segments as the RL agent is given more time to adapt to the new reward model after each update.

### D.3.3. ENSEMBLE OF REWARD FUNCTIONS

As is standard practice in many of our baselines (Lee et al., 2021; Park et al., 2022; White et al., 2024), we learn an ensemble of reward functions rather than a single reward function. Each function is trained independently on the same data provided by the simulated teacher. When providing rewards to the agent, we use the mean of the ensemble's outputs.

### D.3.4. REPLAY BUFFER UPDATE

As in previous preference learning methods (White et al., 2024; Lee et al., 2021; Park et al., 2022; Hu et al., 2024), after each reward update, we relabel all samples in the replay buffer with the newly estimated reward. This technique helps reduce the non-stationarity of the RL task and assists the agent's learning process.

### D.4. Hyperparameters

### D.4.1. SAC HYPERPARAMETERS

**Offline Setting Using SB3 SAC:** We use the default SB3 SAC parameters for the offline experiments.

*Table 10.* Hyperparameters of SB3 SAC

| Hyperparameter | Value | Hyperparameter | Value |
|---|---|---|---|
| Policy | MLP | Critic target update freq | 1 |
| Init temperature | 0.1 | Critic EMA | 0.005 |
| Learning rate | 3e-4 | Discount | 0.99 |
| Batch size | 256 | | |

**Online Setting Using PEBBLE SAC:** We use the default SAC parameters mentioned in (Hu et al., 2024) (see Table D.4.1).

### D.5. Reward Learning Hyperparameters

**Offline Setting:** See the hyperparameters used in Tables 12, 13, and 14.

**Online Setting:** As mentioned in the main text, we collect the initial (first 40) feedback in finer bins. Later, we merge the bins to be coarser. Here, we mention the return ranges the simulated teacher uses to assign trajectories into bins:

*Table 11.* Hyperparameters of PEBBLE SAC

| Hyperparameter | Value | Hyperparameter | Value |
|---|---|---|---|
| Discount | 0.99 | Critic target update freq | 2 |
| Init temperature | 0.1 | Critic EMA | 0.005 |
| Alpha learning rate | 1e-4 | Actor learning rate | 5e-4 (Walker_walk, Cheetah_run) |
| Critic learning rate | 5e-4 (Walker_walk, Cheetah_run) | | 1e-4 (Other tasks) |
| | 1e-4 (Other tasks) | Actor hidden dim | 1024 |
| Critic hidden dim | 1024 | Actor hidden layers | 2 |
| Critic hidden layers | 2 | Batch size | 1024 |
| | | Optimizer | Adam (Kingma & Ba, 2015) |

*Table 12.* Common Hyperparameters

| Hyperparameter | Value | Hyperparameter | Value |
|---|---|---|---|
| $B$ | 64 | Ranking regularization (Blondel et al., 2020) | 1.0 |

`Walker-walk`: {start:[0, 10, 20, 30, 40, 50, 60, 80, 100, 150, 200, 300, 400, 500, 600, 800, 1000], end:[0, 30, 60, 100, 200, 300, 400, 500, 600, 800, 1000] }

`Walker-stand`: {start:[0, 100, 130, 140, 150, 160, 170, 200, 300, 400, 500, 600, 800, 1000], end: [0, 100, 140, 160, 200, 300, 400, 500, 600, 800, 1000]}

`Humanoid-stand`: [0, 0.01, 10, 30, 50, 80, 100, 150, 200, 300, 400, 500, 600, 800, 1000]

`Cheetah-run`, `Quadruped-walk/run`: {[0, 5, 10, 20, 30, 40, 50, 60, 80, 100, 150, 200, 300, 400, 500, 600, 800, 1000], end:[0, 30, 60, 100, 200, 300, 400, 500, 600, 800, 1000]}

See the hyperparameters used in Tables 12, 13, and 15.

*Table 13.* Model Architecture

| Model | # Hidden layers | # Hidden units | Intermediate Activation | Final Activation |
|---|---|---|---|---|
| Medium | 1 | 10 | ReLU (Agarap, 2018) | N/A |
| Large (Offline) | 1 | 100 | ReLU | N/A |
| Large (Online) | 1 | 100 | ReLU | Tanh |

*Table 14.* Offline Hyperparameters

| Environment | # Reward Updates | Model | # Bins |
|---|---|---|---|
| Reacher | 15000 | Medium | 4 |
| InvertedDoublePendulum | 3000 | Medium | 4 |
| HalfCheetah | 1000 | Large (Offline) | 6 |

*Table 15.* Online Hyperparameters

| Hyperparameter | Value | Hyperparameter | Value |
|---|---|---|---|
| Training Steps | 2M (Humanoid, Quadruped), 1M (Other Tasks) | Reward Updates | 500 (Humanoid), 1000 (Others) |
| | | #Preference per session | 20 (Humanoid), 10 (Other tasks) |
| Reward L2 ($\beta$) | 0.005 (Humanoid), 0.01 (Other tasks) | #Bins | Dynamic |
| $\delta$ | 50 | Reward Learning Rate | 3e-4 |
| Model | Large (Online) | Reward Optimizer | Adam |
| Ensamble size | 3 | | |

