# OpenReview forum: "Reward Learning through Ranking Mean Squared Error"
_ICML.cc/2026/Conference — ICML 2026 regular_

### Official Review · Reviewer_b4LN · 2026-03-04

**Soundness:** 3
**Presentation:** 3
**Significance:** 3
**Originality:** 3
**Overall Recommendation:** 4
**Confidence:** 3

**Summary:**

This paper proposes R4, a rating-based reward learning method that uses a ranking mean squared error (rMSE) loss and directly perform optimization on reward model with the help of differentiable ranking models. The paper proposes a theoretical guarantee for its method, and included comprehensive experiments to evaluate the method's effectiveness, data efficiency, robustness against noise and quality of learned reward model. Both simulated and real-human data is included, suggests the method's practical feasibility.

**Compliance With Llm Reviewing Policy:**

Affirmed.

**Final Justification:**

This paper provides a new framework of RL from ranking. The paper proposes a theoretical guarantee for its method, and included comprehensive experiments to evaluate the method's effectiveness, data efficiency, robustness against noise and quality of learned reward model. Both simulated and real-human data is included, suggests the method's practical feasibility. In rebuttal, the authors provides additional justification on theoretical part and cognitive load, as well as uses new experiments to show generalization around ranking operators and tasks. I raise my score from 3 to 4.

**Key Questions For Authors:**

1. Theoretically, how will human irrationality/noise impact the learning performance?
2. Can this method generalize across differentiable ranking operators?
3. Can this method work on more complex systems/tasks, such as robot manipulation? Or on tasks which is harder to rank such as half cheetah back-flipping?

I will raise the score if the authors address the problems.

**Limitations:**

Limitations on theoretical part should be discussed.

**Strengths And Weaknesses:**

## Strengths:
1. The paper is well-written, the idea is straightforward and very easy to understand. Yet it has good performance, successfully leverages the information of human ranking of trajectories.
2. The experiments are comprehensive, covering baseline comparison, ablations and human user study.
3. The paper includes a theoretical analysis to support the proposed method.

## Weaknesses:
1. The theoretical guarantees rely on very strong assumptions. The analysis assumes that human ratings are noise-free and perfectly ordered, meaning that if one trajectory is rated higher than another, its true return under the human’s hidden reward is strictly higher. In practice, human feedback is often inconsistent and noisy.
2. This method depends heavily on a differentiable ranking operator. Its performance may be sensitive to the ranking operator itself. This reliance raises concerns about ability of generalization across different differentiable ranking models.
3. I am not fully convinced that the feedback budget comparison is fair. The paper counts the number of rated trajectories for rating-based methods and the number of pairwise comparisons for preference-based methods, arguing that each comparison involves two trajectories so preferences are less efficient. However, cognitive effort is not necessarily proportional to the number of trajectories shown. A binary preference may be much easier than assigning an absolute rating. Also, ratings with more bins may be more fatigued for human. A more careful discussion of annotation cost or time would make the comparison more convincing.
4. Tasks are relatively simple dm-control tasks, the difficulty of human labeling with more complex system may impact the performance.

---

> ### Author Rebuttal · Authors · 2026-03-30
>
> We thank for the reviewer for their thoughtful feedback!
>
> **W1/Q1 [theoretical guarantees under noisy labels?]**:
>
> We found that our original theoretical guarantees still hold as long as the noise process does not systematically invert the expected ratings.
>
> More specifically, we model noisy ratings as being sampled from a conditional distribution
> $c(\tau) \sim P(\cdot \mid c^{\ast}(\tau))$, where $c^*(\tau)$ is the true noise-free label.
> This captures the realistic setting where human raters may assign different ratings to the same trajectory.
>
> We then analyze the expected loss under this noise model, and show that the minimizer of
> the expected rMSE loss is the conditional expectation of the observed ratings
> $\mu(\tau) = \mathbb{E}[c(\tau) \mid \tau]$ (that is, the average rating a trajectory
> $\tau$ would receive across all possible noisy human ratings). Concretely, the expected
> loss decomposes as $(\mathcal{R}(\tau) - \mu(\tau))^2 + \text{Var}(c(\tau))$, where
> $\mathcal{R}(\tau)$ is the reward model's predicted rating for trajectory $\tau$. The
> variance term is irreducible noise, and the reward model is driven to fit the mean rating
> per trajectory.
>
> This means the our original theoretical guarantees still hold as long as
> the noise process does not systematically invert the expected ratings ( i.e., $c^{\ast}(\tau_{i}) > c^{\ast}(\tau_{j}) \implies \mu(\tau_{i}) > \mu(\tau_{j})$). This is a mild and
> realistic condition, consistent with the variability observed in our user study.
>
> The main failure case would be adversarial or heavily biased ratings, where a poor trajectory consistently receives higher average ratings than a good one,  a scenario unlikely in standard human feedback settings.
>
> **W2/Q2 [Generalization across differentiable ranking operators]**:
> To investigate this concern, we ran an additional experiment in the offline reward learning setting in Reacher using an alternative differentiable ranking operator proposed by [1]. We found no significant difference in performance compared to our default operator [2], suggesting that R4 is not sensitive to the specific choice of differentiable ranking operator and that the method can generalize across different implementations. See Figure 1 in the anonymous link here: https://sites.google.com/view/icml26r4/home
>
> We have added these experiments to the appendix.
>
>
> **W4/Q3 [Generalization to robot manipulation]**:
> We chose dm-control tasks as they are the standard benchmarks for preference- and rating-based RL [3-10].
>
> That said, we took this feedback as an opportunity to evaluate R4 on more complex robotic manipulation tasks from the MetaWorld suite [11]. Specifically, we ran experiments on three tasks: button-press, window-open, and handle-press, comparing R4 against RbRL [4] and PEBBLE [6]. R4 significantly outperforms PEBBLE in two out of three environments and shows significant improvement over RbRL in one. In the remaining cases, R4 performs comparably. These results provide clear evidence that R4’s efficacy generalizes beyond robotic locomotion to complex manipulation settings. See Figure 2 in the anonymous link: https://sites.google.com/view/icml26r4/home
>
> We now include these results in the Appendix.
>
> **W3 [Fairness of feedback budget]:**
> Evidence regarding the cognitive load of ratings versus preferences is currently conflicting (e.g., [3] vs. [4]). We aimed to address this by avoiding a predefined rating scale; by allowing participants to choose their preferred granularity (minimum of 2 bins), our design accommodates the simplest possible cardinal scale while granting participants the autonomy to determine their own rating granularity based on their cognitive comfort.
>
> Despite the availability of this 'low-effort' 2-bin option, all participants voluntarily chose between 4 and 12 bins. This behavior suggests that users not only found the cognitive load of multi-bin ratings to be negligible, but actively preferred the increased expressivity. This is further corroborated by our NASA Task Load Index (TLX) results: participants reported an average overall workload of only 2.05 out of 7, confirming that the rating task was perceived as low-effort and cognitively accessible.
>
> **Referenccs**:
>
> [1] https://arxiv.org/abs/1905.11885
>
> [2] https://arxiv.org/abs/2002.08871
>
> [3] https://arxiv.org/abs/2108.05709
>
> [4] https://arxiv.org/abs/2307.16348
>
> [5] https://arxiv.org/abs/1706.03741
>
> [6] https://arxiv.org/abs/2106.05091
>
> [7] https://arxiv.org/abs/2203.10050
>
> [8] https://arxiv.org/abs/2305.17400
>
> [9] https://arxiv.org/pdf/2502.02921
>
> [10] https://arxiv.org/abs/2405.00746
>
> [11] https://arxiv.org/abs/1910.10897

---

> > ### Author Rebuttal · Reviewer_b4LN · 2026-04-01
> >
> > I am interested to see the new theoretical result. It makes sense that human noise will inject additional variance. The added experiment on different ranking operators and manipulation tasks are good. Empirical results on cognitive load is also provided to address my concern.

---

> > > ### Author Response · Authors · 2026-04-02
> > >
> > > We have added the full proof for the theoretical guarantees under noisy labels to this anonymous link:
> > > https://sites.google.com/view/icml26r4/home
> > >
> > > If we have fully resolved your concerns, we kindly ask that you consider increasing your score.

---

### Official Review · Reviewer_Gfgm · 2026-03-10

**Soundness:** 3
**Presentation:** 4
**Significance:** 3
**Originality:** 3
**Overall Recommendation:** 5
**Confidence:** 3

**Summary:**

he paper studies how to learn reward models for RL with human feedback, aiming to improve over binary preference feedback and prior rating-based methods that do not properly use the ordinal structure of ratings. The authors propose R4, a rank-based mean squared error objective that trains the reward model by enforcing the ordering implied by ratings instead of treating ratings as ordinary classification labels with fixed class boundaries. The method appears meaningfully better than the closest prior rating-based baseline and competitive with several preference-based methods on the tested control tasks.

**Compliance With Llm Reviewing Policy:**

Affirmed.

**Key Questions For Authors:**

NA

**Limitations:**

Yes.

**Strengths And Weaknesses:**

## Strengths
1. The main idea is simple and intuitive, since ordinal ratings should naturally be modeled as ordered rather than as unrelated classes.
2. The paper combines theory, simulated experiments, online training experiments, and a small human study, which gives the work breadth.
3. The experiments have both ideal (ratings from env reward) and practical (ratings from human labelers) settings.

## Weaknesses
1. The comparison with preference-based baselines is not fully fair because ratings and pairwise preferences do not require the same human effort or provide the same amount of information.
2. The experiments are limited to a narrow set of robotic continuous-control tasks, so generalization is unclear. The environments/tasks in this paper are simple and clean as they have well-defined rewards. However, in a more complex environment, where it is almost impossible to give one single well-defined reward (e.g., human preference in LLM's responses or multi-objective reward), it is unclear if this method still outperforms the baselines by a lot, or the gain diminishes.

---

> ### Author Rebuttal · Authors · 2026-03-30
>
> We thank the reviewer for their useful feedback and questions!
>
> **W1 [comparison between preferences and ratings]**:
>
> We agree that ratings and pairwise preferences differ in both information density and possibly the cognitive effort required to provide them; in fact, this disparity was a primary motivation for our work. While a binary preference provides a maximum of 1 bit of information, a rating scale offers significantly more information per label ($\log_2(n)$ bits for $n$ bins). In this paper, we aimed to explore whether rating-based approaches can leverage this higher density to improve reward learning.
>
> To address the concern that ratings might be more cognitively demanding, our study design allowed participants the flexibility to choose their own rating granularity, with a minimum requirement of only two bins. Despite the availability of a "low-effort" two-bin option, all participants voluntarily chose between 4 and 12 bins, suggesting they preferred the increased expressivity and did not find the added granularity to be a deterrent. This is quantitatively supported by our NASA Task Load Index (TLX) results, where participants reported an average overall workload of only 2.05 out of 7. These findings confirm that while ratings provide a richer supervision signal, the cognitive effort can remain low.
>
> **W2 [Limited scope of robotic envs]**:
> We chose dm-control tasks as they are the standard benchmarks for preference- and rating-based RL [1–7]. We also note that several of these tasks themselves have multiple reward components [8, 9]. These tasks, therefore, reflect some of the complexity the reviewer describes.
>
> That said, we took this feedback as an opportunity to evaluate R4 on more complex robotic manipulation tasks from the MetaWorld suite [10]. Specifically, we ran experiments on three tasks: button-press, window-open, and handle-press, comparing R4 against RbRL [7] and PEBBLE [3]. R4 significantly outperforms PEBBLE in two out of three environments and shows significant improvement over RbRL in one. In the remaining cases, R4 performs comparably. These results provide clear evidence that R4’s efficacy generalizes beyond robotic locomotion to complex manipulation settings. See Figure 2 in the anonymous link: https://sites.google.com/view/icml26r4/home
>
> We now include these results in the Appendix.
>
>
> **Referencs**:
>
> [1] https://arxiv.org/abs/1706.03741
>
> [2] https://arxiv.org/abs/2106.05091
>
> [3] https://arxiv.org/abs/2203.10050
>
> [4] https://arxiv.org/abs/2305.17400
>
> [5] https://arxiv.org/pdf/2502.02921
>
> [6] https://arxiv.org/abs/2307.16348
>
> [7] https://arxiv.org/abs/2405.00746
>
> [8] https://github.com/google-deepmind/dm_control/tree/1226983c2ba0f22c3db8ad11824952592c8c2182/dm_control/suite
>
> [9] https://github.com/openai/gym/tree/master/gym/envs/mujoco
>
> [10] https://arxiv.org/abs/1910.10897

---

> > ### Author Rebuttal · Reviewer_Gfgm · 2026-04-01
> >
> > Thank you for the reply. This is an insightful paper. I remain my score.

---

### Official Review · Reviewer_TFWx · 2026-03-12

**Soundness:** 3
**Presentation:** 3
**Significance:** 2
**Originality:** 3
**Overall Recommendation:** 4
**Confidence:** 3

**Summary:**

An important topic in the study of reinforcement learning algorithms is reward design, which is the subject of this paper. It also presents a new method called Ranked Return Regression. It also evaluates this strategy in the gym and DMC suite.

**Compliance With Llm Reviewing Policy:**

Affirmed.

**Final Justification:**

R4 replaces RbRL's class-boundary-dependent cross-entropy with a ranking-based MSE over soft ranks. The completeness and minimality guarantees for the rMSE solution set are the paper's strongest contribution, giving a formal reason to prefer this objective. The human-user study, though small, provides real evidence of robustness under inter-rater variability.

The rebuttal resolved my main concerns. The noise analysis (Q1) shows the rMSE minimizer recovers conditional mean ratings, with guarantees holding as long as expected ratings preserve ordinal structure. The budget clarification (Q2) confirms that equal-trajectory budgets would further advantage R4. The argument that the human study on Reacher and Hopper serves as a more realistic noise test than synthetic flipping (Q3) is reasonable. The within-class variation clarification corrects a misunderstanding on my part: rMSE does not force equal predicted returns within a class, unlike RbRL.

My residual concern is the gap between noise-free theory and noisy practice; formalizing the rebuttal's noise argument would improve the paper. Evaluation is also limited to locomotion tasks. These are limitations, not fatal flaws. I maintain my score of Weak Accept.

**Key Questions For Authors:**

Q1: Assumption 1 calls for noise-free ratings, despite Figure 4 showing that people often rate similar-return trajectories differently. How do the completeness and minimality guarantees change when a fraction eta of the ratings deviate from the ordinal structure?


Q2: The feedback budget in Section 6.2 counts ratings for R4 but employs pairwise comparisons for preference methods. Because a comparison requires the observation of two trajectories, preference methods require twice as many trajectory viewings under the same budget. Did you test with equal numbers of observed trajectories rather than equal numbers of labels?


Q3: The noise robustness test only covers the Inverted Double Pendulum (Figure 24), where R4 and RbRL perform similarly under clean labels. Can you report noise sensitivity on tasks like Half Cheetah, where R4 has a larger clean-label advantage?

**Limitations:**

yes

**Strengths And Weaknesses:**

Strengths: The rMSE loss is elegant in its simplicity. Instead of defining a parametric model over rating classes or establishing class boundaries, R4 streamlines the process to "sort the predicted returns and check if the ranking matches the ratings".

Weaknesses: The theoretical guarantees are based on Assumption 1, which assumes deterministic, noise-free ratings: when i < j, every trajectory in class c_i actually has a lower return than every trajectory in class c_j under the human's implicit reward r*. Only one environment (the Inverted Double Pendulum) was used to validate the noise robustness experiments, which resulted in inadequate coverage; rMSE seems to ignore within-class variation, treating all trajectories with the same class equally.

---

> ### Author Rebuttal · Authors · 2026-03-30
>
> We appreciate the constructive feedback!
>
> **Q1 [theoretical guarantees under noisy labels?]**:
>
> We found that our original theoretical guarantees still hold as long as the noise process does not systematically invert the expected ratings.
>
> More specifically, we model noisy ratings as being sampled from a conditional distribution
> $c(\tau) \sim P(\cdot \mid c^{\ast}(\tau))$, where $c^*(\tau)$ is the true noise-free label.
> This captures the realistic setting where human raters may assign different ratings to the same trajectory.
>
> We then analyze the expected loss under this noise model, and show that the minimizer of
> the expected rMSE loss is the conditional expectation of the observed ratings
> $\mu(\tau) = \mathbb{E}[c(\tau) \mid \tau]$ (that is, the average rating a trajectory
> $\tau$ would receive across all possible noisy human ratings). Concretely, the expected
> loss decomposes as $(\mathcal{R}(\tau) - \mu(\tau))^2 + \text{Var}(c(\tau))$, where
> $\mathcal{R}(\tau)$ is the reward model's predicted rating for trajectory $\tau$. The
> variance term is irreducible noise, and the reward model is driven to fit the mean rating
> per trajectory.
>
> This means the our original theoretical guarantees still hold as long as
> the noise process does not systematically invert the expected ratings ( i.e., $c^{\ast}(\tau_{i}) > c^{\ast}(\tau_{j}) \implies \mu(\tau_{i}) > \mu(\tau_{j})$). This is a mild and
> realistic condition, consistent with the variability observed in our user study.
>
> The main failure case would be adversarial or heavily biased ratings, where a poor trajectory consistently receives higher average ratings than a good one,  a scenario unlikely in standard human feedback settings.
>
> **Q2 [feedback budget, labels vs observed trajectories]**:
> We define the feedback budget based on the total number of labels provided rather than the number of observed trajectories to make it more fair for preference-based RL (PbRL) baselines. If the budget were instead capped by the number of observed trajectories, our method would be further advantaged, as the PbRL budgets would be effectively cut in half. For example, within a budget of 100 observed trajectories, a rating-based approach receives 100 individual labels, whereas a PbRL approach—requiring two trajectories per comparison—would receive only 50 labels.
>
> **Q3 [noise robustness only in Pendulum]**:
> We agree that the simulated noise experiments could be expanded. These experiments were designed to demonstrate that R4 remains functional under high levels of synthetic noise. However, synthetic noise does not fully capture the complexity of real human feedback, which can include systematic biases (e.g., myopic bias) [1].
>
> We believe the most realistic test of noise robustness is our human subject study (Section 6.3), where noise arises naturally from human raters. Human raters introduced variability in two ways: by using different numbers of rating classes, and by assigning different ratings to trajectories with similar environment returns. Under these human noise conditions, R4 consistently outperformed RbRL in two additional environments, Reacher and Hopper, providing stronger evidence of robustness than a synthetic noise test can offer.
>
> **Weakness [MSE seems to ignore within-class variation]**:
> We would like to clarify that while trajectories within the same rating class share the same target label, rMSE does not require their predicted returns to be identical.
> In R4, once a human rates a set of trajectories, one trajectory is sampled from each rating class. The predicted returns for each sampled trajectory are then ranked using a differentiable ranking operator to produce soft ranks. The rMSE loss constrains these soft ranks to match the assigned rating labels — not the returns themselves.
>
> For example, suppose three trajectories are sampled from classes c=[0,1,2]. Predicted returns of [5,15,25] and [5,12,25] would both yield soft ranks [0,1,2] and therefore the same rMSE loss, even though the predicted return for the trajectory in class 1 differs between the two cases.
> This stands in contrast to the RbRL objective [2], which explicitly forces all trajectories within the same rating class to have the same predicted return. rMSE does not impose such equality constraints, allowing the reward model to preserve distinctions between trajectories that share a rating label.
>
> **References:**
>
> [1] https://arxiv.org/abs/2208.10687
>
> [2] https://arxiv.org/abs/2307.16348

---

> > ### Author Rebuttal · Reviewer_TFWx · 2026-04-02
> >
> > I appreciate the authors' comprehensive response and their efforts to clarify the technical details. The provided explanations have resolved the majority of my initial questions. As such, I will keep my score as is.

---

### Official Review · Reviewer_d2sr · 2026-03-14

**Soundness:** 3
**Presentation:** 3
**Significance:** 3
**Originality:** 3
**Overall Recommendation:** 4
**Confidence:** 2

**Summary:**

Reward function design is a major bottleneck in RL. Existing preference-based RL (PbRL) is limited to 1-bit binary comparisons, while rating-based RL (RbRL) uses cross-entropy loss that forces predicted returns to converge to class midpoints, ignoring within-class diversity.

Proposes R4: treats ratings as ordinal feedback, combines differentiable sorting (soft ranks) with MSE loss (rMSE). Eliminates class boundary hyperparameters, supports dynamic rating classes, and allows the reward model to produce continuous, fine-grained reward signals as long as ordering is preserved.

R4 consistently outperforms RbRL and preference-based methods (PEBBLE, SURF, QPA) in offline/online simulated experiments. In a human user study (n=8), R4 significantly outperforms RbRL (p<0.006) despite high inter-participant variability. Provides theoretical guarantees of completeness and minimality for the rMSE solution set.

**Compliance With Llm Reviewing Policy:**

Affirmed.

**Final Justification:**

The authors addressed my concerns satisfactorily in the rebuttal. I maintain my score.

**Key Questions For Authors:**

- The theoretical guarantees (completeness, minimality) assume noise-free ratings (Assumption 1). How are these guarantees affected in realistic settings where humans assign different ratings to trajectories with similar returns, as observed in the user study?
- Are there evaluations of reward model quality using criteria independent of the environment reward (e.g., human ranking agreement, trajectory quality surveys)?

**Limitations:**

Circular evaluation logic: the goal is to learn rewards without environment reward, but learned reward quality is evaluated only by environment reward. The practical need for this method is weakened in benchmarks where the environment reward is already sufficient

**Strengths And Weaknesses:**

**Strengths**
- Clear theoretical contribution: proves rMSE solution set is complete and minimal, while RbRL may exclude the true reward function (Proposition 1, Theorems 1 & 2)
- Systematic experimental design: progressively increases realism through offline (simulated) → online (simulated) → human user study
- Includes actual human participant experiments and promises to release the dataset

**Weaknesses**
- The claim that rating is less cognitively demanding than preference comparison is based on a small study (n=8) in simple robotic environments, limiting generalizability
- Allowing participants to freely choose the number of rating classes may create conditions favorable to R4 and unfavorable to RbRL (which is sensitive to bin count). A controlled comparison with fixed rating classes would have made the results easier to interpret

---

> ### Author Rebuttal · Authors · 2026-03-30
>
> We thank the reviewer for their thoughtful feedback!
>
> **W1 [sample size]:**
> While these initial results offer a compelling proof-of-concept regarding the low cognitive load of ratings, we agree that larger-scale validation is necessary. We have added this to our future work section of the paper.
>
> **W2 [freely choosing # of rating classes]**:
> We deliberately allowed participants to select their own bin counts. We believe that if an algorithm requires a specific, fixed bin count to remain competitive—especially if that count is cognitively taxing for a human—its real-world applicability is compromised. By providing flexibility, we ensured that the feedback collected remained within the threshold of what participants could reliably and accurately provide without inducing cognitive fatigue.
>
> Furthermore, the individual results in Appendix B (Figures 9–21) provide a built-in sensitivity analysis. These results demonstrate that across the entire user-selected range (4 to 12 bins), R4 consistently performed comparably to, or outperformed, RbRL. This suggests that R4 is significantly more robust to a varying number of ratings bins than the baseline, which the reviewer correctly notes can be sensitive to bin count.
>
> **Q1 [theoretical guarantees under noisy labels]**:
> We found that our original theoretical guarantees still hold as long as the noise process does not systematically invert the expected ratings.
>
> More specifically, we model noisy ratings as being sampled from a conditional distribution
> $c(\tau) \sim P(\cdot \mid c^{\ast}(\tau))$, where $c^*(\tau)$ is the true noise-free label.
> This captures the realistic setting where human raters may assign different ratings to the same trajectory.
>
> We then analyze the expected loss under this noise model, and show that the minimizer of
> the expected rMSE loss is the conditional expectation of the observed ratings
> $\mu(\tau) = \mathbb{E}[c(\tau) \mid \tau]$ (that is, the average rating a trajectory
> $\tau$ would receive across all possible noisy human ratings). Concretely, the expected
> loss decomposes as $(\mathcal{R}(\tau) - \mu(\tau))^2 + \text{Var}(c(\tau))$, where
> $\mathcal{R}(\tau)$ is the reward model's predicted rating for trajectory $\tau$. The
> variance term is irreducible noise, and the reward model is driven to fit the mean rating
> per trajectory.
>
> This means the our original theoretical guarantees still hold as long as
> the noise process does not systematically invert the expected ratings ( i.e., $c^{\ast}(\tau_{i}) > c^{\ast}(\tau_{j}) \implies \mu(\tau_{i}) > \mu(\tau_{j})$). This is a mild and
> realistic condition, consistent with the variability observed in our user study.
>
> The main failure case would be adversarial or heavily biased ratings, where a poor trajectory consistently receives higher average ratings than a good one,  a scenario unlikely in standard human feedback settings.
>
> **Q2 [reward evaluations independent of env reward?]:**
> We used the Trajectory Alignment Coefficient (TAC) [1] as a means to evaluate the quality of the reward model. This metric in itself does not rely on having an environmental reward. It measures the similarity in trajectory preferences/rankings from a reward model and a human. The reward model quality analysis we report in Section 6.4 is based on our simulated human experiments. In these simulated human experiments, the environment reward function was used to provide the simulated ''human'' ratings/preferences. We have updated our paper to make it clearer that this is why we used TAC to compare the learned reward model preferences with the simulated human preferences.
>
> **Limitation 1 [Circular evaluation logic:]**
> The training and evaluation scheme — learning a reward function from preferences or ratings derived from an environment reward, then evaluating performance against that same reward — is standard practice across preference- and rating-based RL algorithms [2-8]. We follow this convention to (1) enable direct comparison against prior methods and (2) facilitate experiments across a large set of environments. We agree, however, that this is a limitation of benchmarking in this field.
>
> That said, this was one of our primary motivations for conducting a human subject study: to provide evidence that R4 can learn effective reward functions from actual human ratings, independent of a simulated preference oracle. Such studies are uncommon in this field [8].
> We agree that future evaluation on environments without a natural reward function, such as cheetah jumping or other expressive locomotion tasks, would further validate the approach, and we now call this out in the future work section of our paper.
>
> **References:**
> [1] https://arxiv.org/abs/2503.05996
> [2] https://arxiv.org/abs/1706.03741
> [3] https://arxiv.org/abs/2106.05091
> [4] https://arxiv.org/abs/2203.10050
> [5] https://arxiv.org/abs/2305.17400
> [6] https://arxiv.org/pdf/2502.02921
> [7] https://arxiv.org/abs/2307.16348
> [8] https://arxiv.org/abs/2405.00746

---

> > ### Author Rebuttal · Reviewer_d2sr · 2026-04-06
> >
> > The authors have addressed my concerns satisfactorily. No further questions.

---

### Decision · Program_Chairs · 2026-04-30

**Decision:**

Accept (regular)

**Comment:**

The reviewers generally found the core idea of R4 interesting and technically sound, highlighting its elegant ordinal formulation, supporting theory, and solid empirical breadth. The main concerns centered on the strong assumptions behind the theory, the fairness of comparisons to preference-based baselines, and the limited scope of the evaluation. In rebuttal, the authors clarified the noisy-label setting, strengthened the discussion of feedback cost, and added additional evidence on generalization and robustness. These responses addressed most reviewer concerns, and the overall reception remained positive.